# ORP5 localizes to ER–lipid droplet contacts and regulates the level of PI(4)P on lipid droplets

Ximing Du[1]* , Linkang Zhou[2]* , Yvette Celine Aw[1] , Hoi Yin Mak[1], Yanqing Xu[1], James Rae[3], Wenmin Wang[2], Armella Zadoorian[1], Sarah E. Hancock[4], Brenna Osborne[4] , Xiang Chen[5], Jia-Wei Wu[5], Nigel Turner[4], Robert G. Parton[3] , Peng Li[2] , and Hongyuan Yang[1]

Lipid droplets (LDs) are evolutionarily conserved organelles that play important roles in cellular metabolism. Each LD is enclosed by a monolayer of phospholipids, distinct from bilayer membranes. During LD biogenesis and growth, this monolayer of lipids expands by acquiring phospholipids from the endoplasmic reticulum (ER) through nonvesicular mechanisms. Here, in a mini-screen, we find that ORP5, an integral membrane protein of the ER, can localize to ER–LD contact sites upon oleate loading. ORP5 interacts with LDs through its ligand-binding domain, and ORP5 deficiency enhances neutral lipid synthesis and increases the size of LDs. Importantly, there is significantly more phosphatidylinositol-4-phosphate (PI(4)P) and less phosphatidylserine (PS) on LDs in ORP5-deficient cells than in normal cells. The increased presence of PI(4)P on LDs in ORP5-deficient cells requires phosphatidylinositol 4-kinase 2-α. Our results thus demonstrate the existence of PI(4)P on LDs and suggest that LD-associated PI(4)P may be primarily used by ORP5 to deliver PS to LDs.

## Introduction

Lipid droplets (LDs) are intracellular organelles that are the primary sites for storing excess lipids (Bersuker et al., 2018; Farese and Walther, 2009; Gao et al., 2019; Olzmann and Carvalho, 2019; Yang et al., 2012). LDs are also involved in several other cellular activities including gene expression regulation, intracellular lipid and membrane trafficking, viral replication, and inflammatory responses. Each LD contains a hydrophobic neutral lipid core of triacylglycerols (TAGs) and cholesteryl esters, enclosed by a monolayer of amphipathic lipids. The surface of LDs is unique among cellular organelles because of its monolayer nature as opposed to bilayer membranes. Importantly, a large number of proteins are attached to the surface of LDs, and these LD-associated proteins often play crucial roles in cellular metabolism. The composition of LD surface lipids impacts LD biogenesis and growth (Fei et al., 2011; Gao et al., 2019) and dictates the targeting and function of LD-associated proteins. For instance, FSP27 plays a critical role in the formation of unilocular LDs (Gong et al., 2011; Nishino et al., 2008), and the amount of phosphatidic acid on the LD surface was reported to impact FSP27 function in LD growth/fusion (Barneda et al., 2015). Thus, the composition of LD surface monolayer is crucial to LD function.

LDs grow in size in the presence of excess neutral lipids; therefore, the surface monolayer must expand accordingly. How LDs acquire surface lipids and maintain their proper composition is a fundamental question in LD biology. Past work demonstrated that key enzymes in phospholipid biosynthesis could translocate to the LD surface during LD growth (Krahmer et al., 2011). It has also been known for a long time that enzymes in sterol biosynthesis can localize to the LD surface. Thus, local lipid synthesis may play a role to supply surface polar lipids. Alternatively, LDs may acquire surface lipids directly from other organelles, especially the ER, which is known to form contact sites with the LDs (Xu et al., 2018). Since there is no vesicular transport between the ER and LDs, lipid transfer proteins (LTPs) would be required to deliver the lipids from the ER to LDs (Wong et al., 2019). Currently, it remains unclear whether and how lipid transfer takes place between the ER and LDs. Finally, phospholipids could also diffuse from ER to LDs when LDs remain physically connected to the ER.

The oxysterol binding protein (OSBP) and its related proteins (OSBP related protein [ORP]) have emerged as important cellular LTPs (Antonny et al., 2018; Du et al., 2015; Olkkonen and Li, 2013; Suchanek et al., 2007; Yang, 2006). There are 12 OSBP/ORP members in humans and 7 members (Osh1–7) in the

[1]School of Biotechnology and Biomolecular Sciences, The University of New South Wales, Sydney, Australia; [2]State Key Laboratory of Membrane Biology and Tsinghua-Peking Center for Life Sciences, School of Life Sciences, Tsinghua University, Beijing, China; [3]Centre for Microscopy and Microanalysis, Institute of Molecular Bioscience, University of Queensland, St. Lucia, Australia; [4]School of Medical Sciences, The University of New South Wales, Sydney, Australia; [5]Institute of Molecular Enzymology, Soochow University, Suzhou, Jiangsu, China.

*X. Du and L. Zhou contributed equally to this paper; Correspondence to Ximing Du: x.r.du@unsw.edu.au; Peng Li: li-peng@mail.tsinghua.edu.cn; Hongyuan Yang: h.rob.yang@unsw.edu.au.

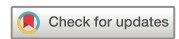

budding yeast *Saccharomyces cerevisiae* (Beh et al., 2001). These proteins all share a conserved ~400-aa OSBP related domain (ORD) found at the C-terminus of OSBP, which has been shown to bind and transfer lipids. Members of the OSBP/ORP family vary in length: the short ORPs comprise primarily the ORD, whereas the long ones possess additional functional domains including a Pleckstrin homology (PH) domain and an FFAT (diphenylalanine in an acidic tract) motif for membrane targeting. Through the PH domain and FFAT motif, some ORPs could simultaneously bind two membranes, promoting the formation of membrane contact sites (Mesmin et al., 2013a). Importantly, it is now well established that ORPs engage phosphatidylinositol 4-phosphate (PI(4)P) to transfer more common lipids such as sterols and phosphatidylserine (PS; Chung et al., 2015; de Saint-Jean et al., 2011; Im et al., 2005; Moser von Filseck et al., 2015). For instance, OSBP delivers cholesterol from the ER to the Golgi and brings back PI(4)P, which is hydrolyzed by the ER-resident phosphatase Sac1. The hydrolysis of PI(4)P supplies energy for OSBP to transfer cholesterol against a concentration gradient (de Saint-Jean et al., 2011; Mesmin et al., 2013b). Recent studies demonstrated that ORPs can also use phosphatidylinositol 4,5-bisphosphate (PI(4,5)P$_2$) to drive the transport of common lipids such as cholesterol and PS (Ghai et al., 2017; Wang et al., 2019).

We hypothesized that some of the OSBP/ORPs might function to deliver lipids to LDs. Here, we performed a localization screen to identify OSBP/ORPs on LDs. We found that ORP5, a protein implicated in the transfer of PS from the ER to the plasma membrane (PM; Chung et al., 2015; Ghai et al., 2017; Moser von Filseck et al., 2015; Sohn et al., 2018), also localized to ER–LD contact sites upon oleate loading. We further identified PI(4)P on the LD surface and demonstrated that ORP5 regulated the presence of PI(4)P on LDs in a phosphatidylinositol 4-kinase 2-α (PI4K2A)–dependent manner.

## Results

### ORP5 specifically localizes to the LD surface

To examine the potential role of OSBP/ORPs in regulating LD dynamics, we performed a mini-scale localization screen. We transfected HeLa cells with cDNAs encoding mCherry-tagged OSBP/ORPs and treated cells with oleate to induce LD formation. Among all ORPs examined, only mCherry-ORP5 displayed strong LD association, as revealed by fluorescence microscopy (Fig. S1). The strong LD association of GFP-tagged ORP5, as well as its naturally occurring isoform B, which misses part of the PH domain (Ghai et al., 2017), was also found in a few other cell lines (Fig. S2). ORP2 could be found on LDs as previously described (Hynynen et al., 2009), but its LD association was much weaker than that of ORP5 (Fig. S1). ORP5 and ORP2 could colocalize on the LD surface in Huh7 cells (Fig. S3 A, top). 22R-hydroxycholesterol, which blocked the LD localization of ORP2 (Hynynen et al., 2009), had no effect on the LD localization of ORP5 (Fig. S3 A, bottom). Similarly, 25-hydroxycholesterol and itraconazole, both of which affect the localization of OSBP (Charman et al., 2017; Strating et al., 2015), also had little impact on the targeting of ORP5 to LDs (Fig. S3 B).

To further characterize the LD targeting of ORP5, we carefully compared ORP5 with ORP8, since they belong to the same subgroup of the ORP family and have a similar domain structure (Fig. 1 A). Confocal microscopy confirmed that ORP5, but not ORP8, specifically localizes to the LD surface (Fig. 1 B). ORP5 and ORP8 have been shown to function at the ER–PM contact sites, where they act as PS transporters and regulate PM PI(4)P and PI(4,5)P$_2$ homeostasis (Chung et al., 2015; Ghai et al., 2017; Sohn et al., 2018). Colocalization studies indicated that, among all cells expressing mCherry-tagged ORP5 or ORP8, the majority of ORP5 (50–75% of transfected cells) localized to both ER–PM contact sites (MAPPER as an ER–PM junction marker; Chang et al., 2013) and LDs after oleate treatment (Fig. 1, C–E). By contrast, ORP8 mainly showed a tubular ER localization under the same growth conditions (Fig. 1, C–E). We also examined the targeting of GFP-ORP5 to LDs by FRAP. The recovery of GFP signal on LDs occurred within 90 s (Fig. 1, F and G). The incomplete recovery of the GFP signals 5 min after photobleaching may be due to the presence of an immobile fraction of GFP-ORP5 anchored to the ER membranes (Fig. 1 G). The targeting of GFP-ORP5 to LDs was further illustrated by 3D reconstruction of the LD surface from the Airyscan confocal images of LDs enveloped by GFP-ORP5 (Fig. 1 H).

### ORP5 localizes to ER–LD contact sites

To further confirm the LD localization of ORP5, we detected GFP-ORP5 by EM in oleate-treated cells. When coexpressed with GFP-binding peptide (GBP) tagged with APEX (soybean ascorbate peroxidase) and in the presence of DAB and H$_2$O$_2$, APEX from the resultant protein complex APEX-GBP/GFP-ORP5 catalyzes the production of an electron-dense substance (Ariotti et al., 2015), which can be visualized by EM to reveal GFP-ORP5 localization (Fig. 2 A). As expected, electron-dense products were seen on isolated PM patches in cells cotransfected with GFP-ORP5 and APEX-GBP (Fig. 2 B, arrowheads), indicating ORP5 localization to ER–PM contact sites as previously reported (Chung et al., 2015; Ghai et al., 2017; Sohn et al., 2018). Strikingly, strong GFP-ORP5 positive signals were revealed around spherical LDs (Fig. 2 B, inlay), confirming ORP5 localization to LD surface. As a negative control, GFP-ORP8–positive signals were observed in the ER but not on LDs (Fig. 2 B, inlay) consistent with our confocal microscopic observations (Fig. 1, B–D).

The association of an ER-anchored ORP5 with LDs suggests a possible ER–LD contact localization of ORP5 (Fig. 2 C). Indeed, the APEX EM experiments also demonstrated that signals positive for GFP-tagged ORP5 and its naturally occurring isoform B (ORP5B; Ghai et al., 2017) were enriched in the contact sites connecting LDs and the ER (Fig. 2 D). Moreover, Airyscan confocal microscopy in Cos-7 cells treated with oleate showed that some of the GFP-ORP5B puncta localized to distinct ER regions closely adjacent to LDs (Fig. 2 E). We also performed immunofluorescence staining and examined the localization of endogenous ORP5 in oleate-treated cells. Due to a lack of high-quality antisera against ORP5, the specificity of the immunofluorescence was verified by treating cells with siRNAs to deplete endogenous ORP5 (Fig. 2, F and G). In ORP5 knockdown cells, the

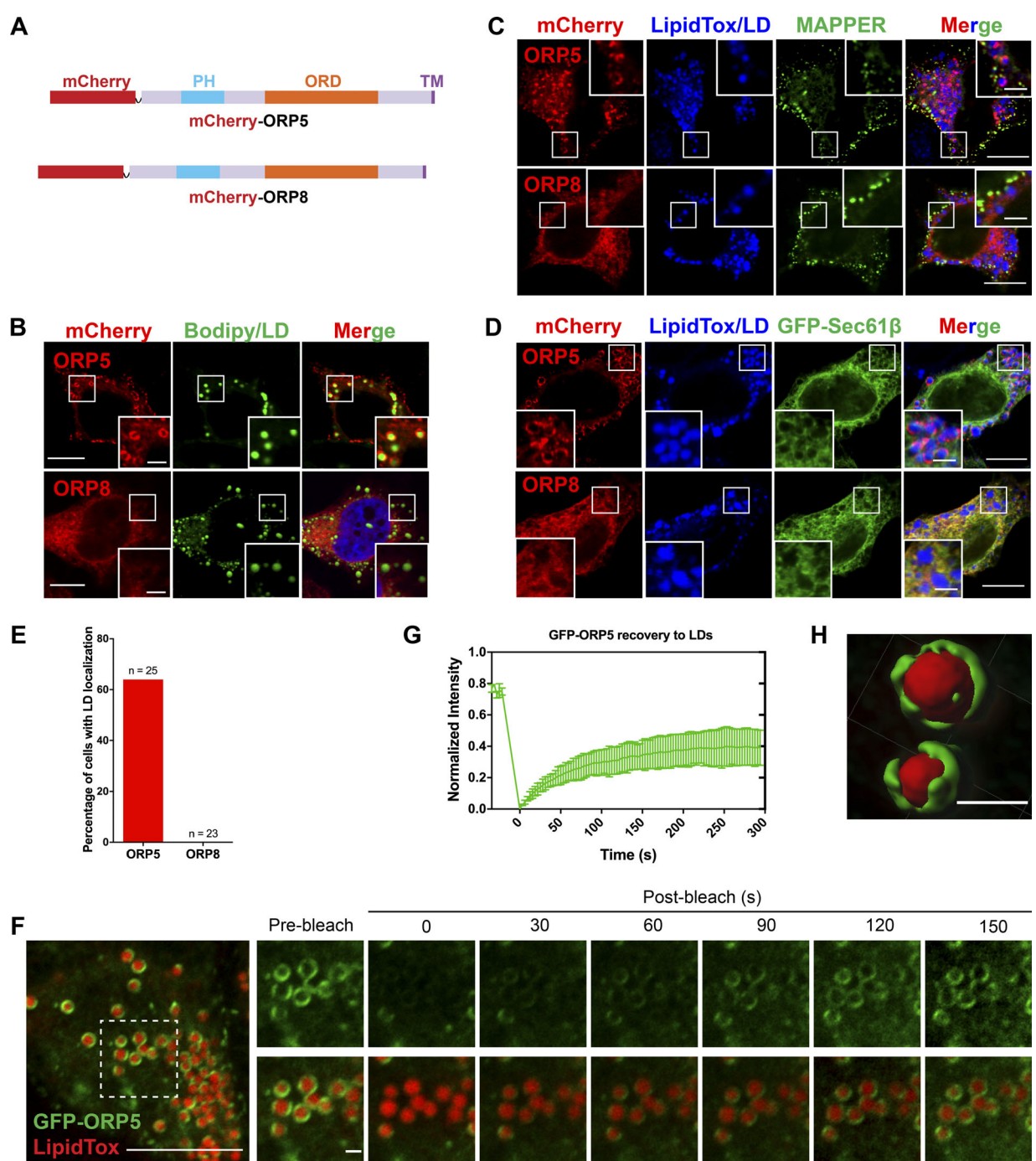

Figure 1.   **ORP5 localizes to LDs. (A)** Diagrams of domain structures for ORP5 and ORP8. **(B)** Confocal images of mCherry-ORP5/8 localizations in HeLa cells treated with oleate for 16 h. LDs were stained with BODIPY. Bars = 10 µm (insets, 2.5 µm). **(C and D)** Confocal images of mCherry-ORP5 or mCherryORP8 localizations in oleate-treated cells transfected with either the ER–PM junction marker MAPPER or the ER marker GFP-Sec61β. LDs were stained with LipiTox DeepRed. Bars = 10 µm (insets, 2.5 µm). **(E)** Percentage of the cells expressing mCherry-tagged ORP5 or ORP8 that show LD localization. **(F)** GFP signal recovery after photobleaching in GFP-ORP5–expressing HeLa cells treated with oleate for 16 h. LDs were stained with LipidTox DeepRed. Bar = 10 µm (inlay, 1 µm). **(G)** Time-dependent recovery of GFP signal in F (n = 2). Mean ± SD. **(H)** 3D reconstruction of LDs enveloped by GFP-ORP5. Bar = 1 µm. All data are representative of at least three independent experiments with similar results.

immunofluorescence signals were barely detectable, but in control siRNA–treated cells, the endogenous ORP5 displayed a punctate localization pattern reminiscent of the ER, indicating that ORP5 may target to specific microdomains of ER membrane upon oleate treatment (Fig. 2 G). Interestingly, some of these ORP5 puncta were found next to LDs, demonstrating that ER-localized ORP5 can make contact with LDs (Fig. 2 G, inlay). Thus, when cells are induced to form LDs, both overexpressed and endogenous ORP5 can localize to the ER–LD contact sites.

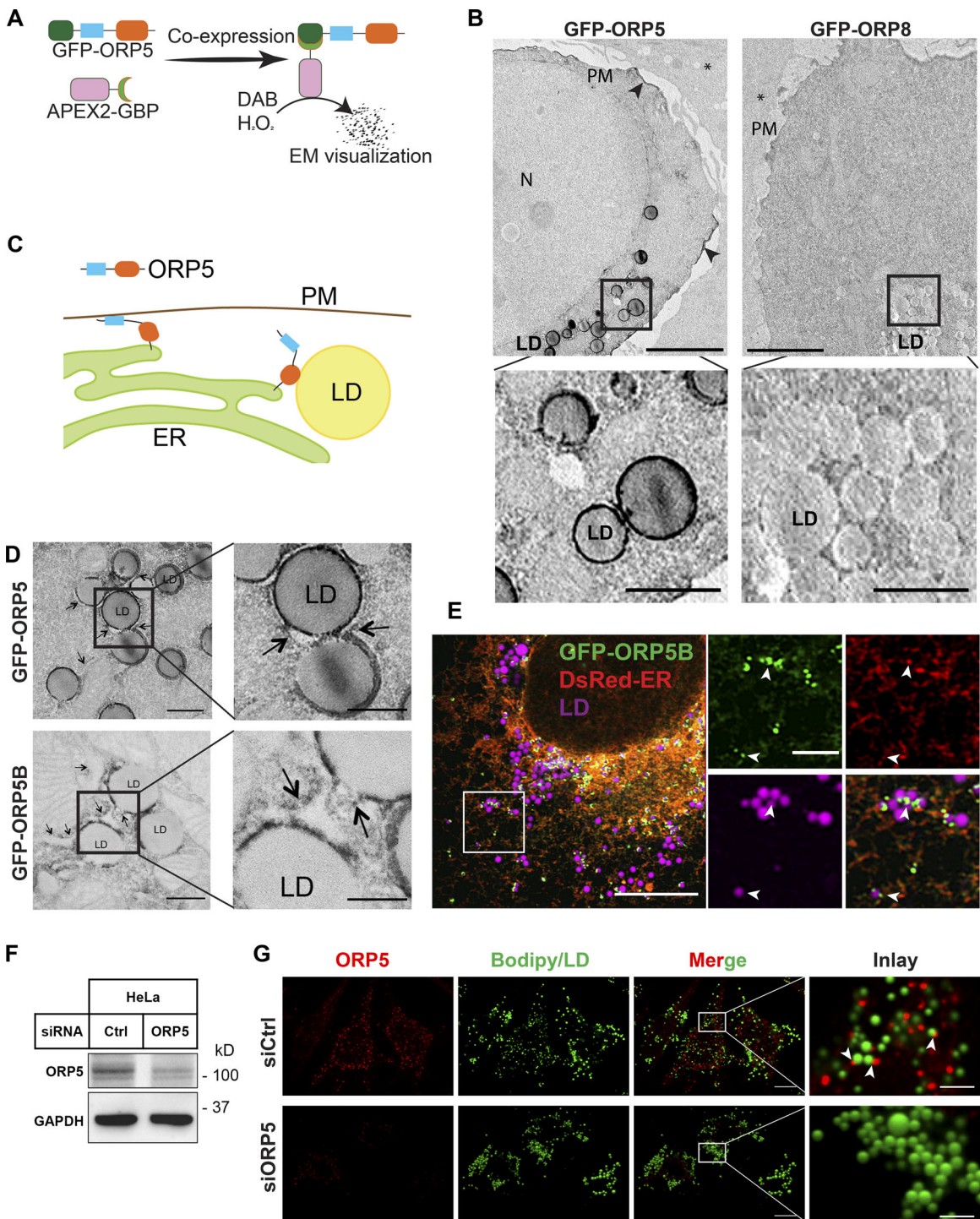

Figure 2. **ORP5 localizes to ER–LD contacts. (A)** Principle of APEX-EM. **(B)** APEX-EM images of oleate-treated cells expressing GFP-ORP5 or GFP-ORP8 together with APEX2-GBP. Arrowheads indicate GFP-ORP5 patch at ER–PM contact sites. N, nucleus. Bars = 10 μm (magnifications, 2.5 μm). **(C)** Cartoon of ORP5 localization at ER–PM and ER–LD contact sites. **(D)** AMEX-EM images of GFP-ORP5 and GFP-ORP5B localization at ER–LD contact sites. Arrows indicate ER. Bars = 0.5 μm (magnifications, 0.25 μm). **(E)** Airyscan confocal image of GFP-ORP5B/DsRed-ER–expressing HeLa cells treated with oleate for 16 h. LDs were stained with LipidTox DeepRed. Arrowheads indicate GFP-ORP5B association with ER and LDs. Bar = 10 μm (inlay, 2.5 μm). **(F)** Western blot analysis of ORP5 in HeLa cells treated with control or ORP5 specific siRNAs. **(G)** Immunofluorescence of endogenous ORP5 in siRNA transfected cells treated with oleate for 16 h. LDs were stained with BODIPY. Arrowheads indicate ORP5 association with LDs. Bars = 10 μm (inlay, 2.5 μm). All data are representative of at least three independent experiments with similar results.

## ORP5-ORD is required for LD targeting

Apart from its C-terminal transmembrane domain, ORP5 has an N-terminal PH domain and a conserved ORD, which are responsible for membrane targeting and lipid transfer, respectively (Chung et al., 2015; Ghai et al., 2017; Sohn et al., 2018). To delineate the domain requirement of ORP5 for LD targeting, we examined the localizations of WT and truncated forms of ORP5 lacking the PH (ORP5ΔPH), ORD (ORP5ΔORD), both PH and ORD (ORP5ΔPHΔORD), or C-terminal transmembrane (ORP5ΔTM) domains. The localization of ORP5B that bears an incomplete PH domain was also examined (Fig. 3 A). Consistent with Fig. 1 (A–E), the majority of WT ORP5 localized to both ER–PM contact sites and LD surfaces in oleate-treated cells (Fig. 3, B and C). As shown in previous studies (Ghai et al., 2017), loss of the PH domain (GFP-ORP5ΔPH and GFP-ORP5B) abolished PM targeting of ORP5; interestingly, this loss favored LD targeting of ORP5 (Fig. 3, B and C). On the other hand, loss of the ORD domain completely abolished LD localization of ORP5 (Fig. 3, B and C). GFP-ORP5ΔORD was mainly found at ER–PM contacts, and GFP-ORP5ΔPHΔORD appeared to be completely trapped in tubular ER, consistent with the requirement of the PH domain for PM targeting and the ORD domain for LD targeting (Fig. 3, B and C). The ORP5ΔTM mutant was enriched on the PM due to the presence of the PH domain. However, it could still localize to the LDs that were close to the PM, indicating that the TM is not essential for the LD targeting of ORP5 (Fig. 3, B and C). We also performed fractionation studies in cells expressing GFP-tagged ADRP, ORP5, ORP5B, and ORP5ΔORD. ADRP (also known as perilipin 2) is a prominent marker that coats the surface of LDs (Brasaemle et al., 1997). Like ADRP, both ORP5 and ORP5B were found in the LD fractions (Fig. 3, D [lane 9–11] and E). In contrast, the mutant ORP5ΔORD is absent from the LD fractions (Fig. 3, D [lane 12] and E). Together, these data demonstrate that the ORD, but not the PH domain, of ORP5 is required for the targeting of ORP5 to LDs.

## An amphipathic helix (AH) within ORP5-ORD appears crucial for LD targeting

Next, we investigated how ORP5-ORD targets ORP5 to LD surface. The secondary structure of ORP5-ORD possesses several α-helices (Ghai et al., 2017). A helical wheel representation of one of these α-helices (aa 422–439) indicates a putative AH within ORP5-ORD (Fig. 4 A). AH domains are known to target proteins to the LD surface (Prevost et al., 2018). We therefore determined if the AH within ORP5-ORD is required for ORP5 targeting to LDs. For this purpose, we constructed two mutants of ORP5, one without the AH (GFP-ORP5ΔAH) and one with key residues of the AH being replaced by cationic arginine (GFP-ORP5-M426R/V429R/L430R or GFP-ORP5-MVL/RRR; Fig. 4 B). Compared with WT GFP-ORP5, both mutants completely lost LD localization (Fig. 4, C and E). We also made similar mutants in GFP-ORP5B, which had stronger LD association than GFP-ORP5 (Fig. 3, B and C). As expected, GFP-ORP5B without the AH or with the AH-disrupting mutations (MVL/RRR) completely lost LD localization (Fig. 4, D and E). These data demonstrate that LD targeting of ORP5 may be driven by the presence of this AH within ORP5-ORD. Interestingly, a similar AH domain was also

identified within ORP8ORD (aa 481–498; Fig. S3 C). Since this AH was not sufficient to direct ORP8 targeting to LDs (Figs. 1 and S3 D), we replaced it with the AH of the ORP5-ORD and constructed a chimeric ORP8 harboring ORP5AH (Fig. S3 E). However, ORP8 with ORP5AH failed to show LD localization (Fig. S3 F). We have also fused the ORP5AH (aa 422–439) domain with GFP and tested its localization in cells treated with oleate. This fusion protein failed to show a clear LD pattern (not depicted). Interestingly, however, when the same GFP-ORP5AH was anchored to the ER by the transmembrane region of ORP5, it could now localize to LDs (Fig. 4, F–H). While these results imply that the AH may be necessary and sufficient for LD targeting of ORP5, this AH region may also be required to maintain the overall fold of the ORD.

## ORP5 regulates LD size

The targeting of ORP5 to the LD surface prompted us to investigate its function at ER–LD contacts. We knocked down ORP5 by RNAi in WT cells using two ORP5-specific siRNAs and examined the LDs (Fig. 5 A). ORP5 silencing increased LD sizes, with the number of small LDs decreased and that of big LDs, especially those with a diameter >2 μm, increased (Fig. 5, B–D). We then used the CRISPR/Cas9 system and knocked out the ORP5 gene (Fig. 5 E). Similar to ORP5 knockdown cells, the population of larger LDs in ORP5 knockout (KO) cells substantially increased (Fig. 5, F–H). Importantly, the phenotype of bigger LDs in ORP5 KO cells was reversed in those cells expressing GFP-ORP5 (Fig. 5, I–K). In contrast, ORP5 mutants deficient for PS (L389D) transport or PI(4)P/PI(4,5)P$_2$ binding (H478A/H479A; Chung et al., 2015; de Saint-Jean et al., 2011; Du et al., 2018; Galmes et al., 2016; Ghai et al., 2017; Maeda et al., 2013; Sohn et al., 2018) failed to reduce the size of LDs in ORP5 KO cells (Fig. 5, L and M), although L389D mutant seemed to have reduced the sizes of the LD population with a diameter >2 μm (Fig. 5 N). Moreover, both L389D and H478A/H479A mutants of ORP5 were mainly enriched at ER–PM contacts. Notably, L389D and H478A/479A mutants were rarely targeted to the LD surface in oleate-treated cells (Fig. 5, L and O). The AH mutant, GFP-ORP5-MVL/RRR, also failed to rescue the enlarged LD phenotype in ORP5 KO (Fig. S4, A–C). These data suggest that the lipid transfer activity of ORP5 is important to its targeting and function at LDs. We also examined the role of ORP5 in the early stages of LD biogenesis using Hpos, which contains the hydrophobic domain of associated with lipid droplet protein 1 fused with the last 20 residues of caveolin-1 (Kassan et al., 2013). Hpos is a sensitive probe that can detect the earliest LDs, whereas lipophilic dyes such as LipidTox can stain only LDs of a certain size. Early LDs (Hpos-positive puncta) became LipidTox positive within 5 min in both ORP5-deficient and control cells (Fig. S4 D, arrowheads), suggesting that ORP5 does not regulate early LD maturation. At a later stage (4 h after oleate treatment), there was a significant increase of TAG content in ORP5-deficient cells compared with WT cells (Fig. S5, A and B). Interestingly, upon 4 h of oleate treatment, the TAG synthesizing enzyme diacylglycerol O-acyltransferase 2 (DGAT2), but not DGAT1, became more frequently associated with LDs in ORP5-deficient cells than in WT cells (Fig. S5, C–F).

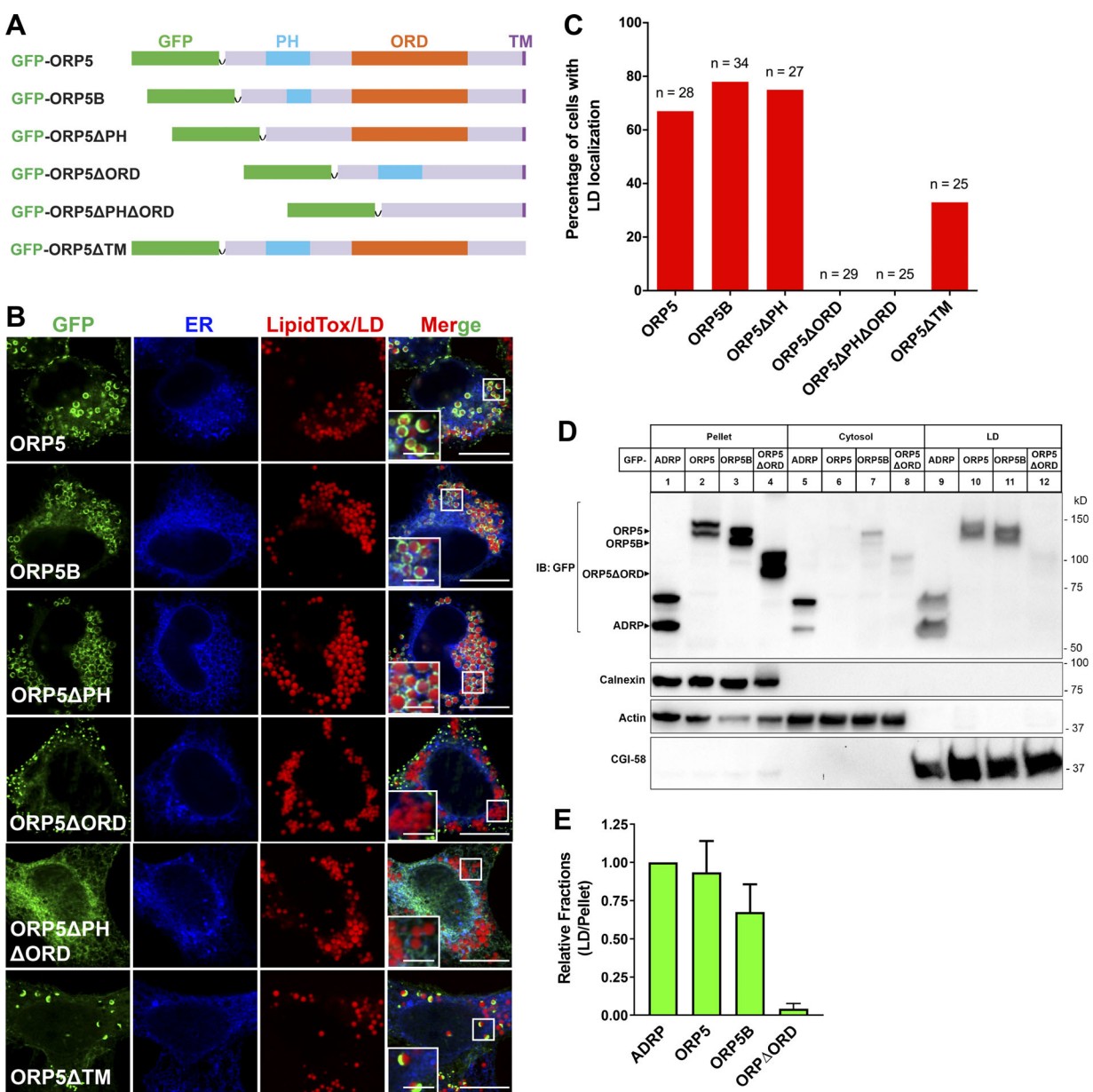

Figure 3. **The ORD of ORP5 is required for its LD targeting. (A)** Constructs of GFP-ORP5 and its variants with domain deletions. **(B)** Confocal images showing the localizations of GFP-ORP5 and variants with domain deletion in cells cotransfected with the ER marker DsRed-ER and treated with oleate for 16 h. LDs were stained with LipidTox DeepRed. Bars = 10 µm (insets, 2.5 µm). **(C)** Percentage of the cells expressing GFP-ORP5 and variants with domain deletion that show LD localizations. **(D)** LD fractionation in cells expressing GFP-ADRP, GFP-ORP5, GFP-ORP5B, and GFP-ORP5ΔORD. IB, immunoblot. **(E)** Relative fractions of LD-associated GFP-ADRP, ORP5, ORP5B, and ORP5ΔPHΔORD (LD/Pellet) in D. Mean ± SD. All data are representative of at least three independent experiments with similar results.

## ORP5 negatively regulates PI(4)P association with LDs

ORP5 has been recently shown as a PS/phosphoinositides exchanger between the ER and the PM (Chung et al., 2015; Ghai et al., 2017; Sohn et al., 2018). This activity of ORP5 is important in maintaining the levels of phosphoinositides such as PI(4)P and PI(4,5)P$_2$ on the PM (Ghai et al., 2017; Sohn et al., 2018). We thus investigated the intracellular distribution of PI(4)P and PI(4,5)P$_2$ by immunofluorescence in WT and ORP5-deficient cells treated with oleate (Hammond et al., 2009). In WT cells, only a very small number of LDs were associated with PI(4)P puncta, whereas in ORP5 KO cells, PI(4)P appeared to be more

dispersed, and a relatively larger number of LDs were found to be PI(4)P positive (Fig. 6, A and B). On the other hand, PI(4,5)P$_2$ signals showed primarily PM localization, and very few LDs appeared to be PI(4,5)P$_2$ positive in both WT and ORP5 KO cells (Fig. 6, A and B).

Besides immunofluorescence, we also used probes to examine the distribution of intracellular PI(4)P and PI(4,5)P$_2$. These probes (GFP-P4M-SidM for PI(4)P [Hammond et al., 2014] and GFP-PLCPH for PI(4,5)P$_2$ [Stauffer et al., 1998]) were transiently expressed in WT and ORP5 KO cells treated with oleate. While mainly displaying dispersed localization, the PI(4)P probe

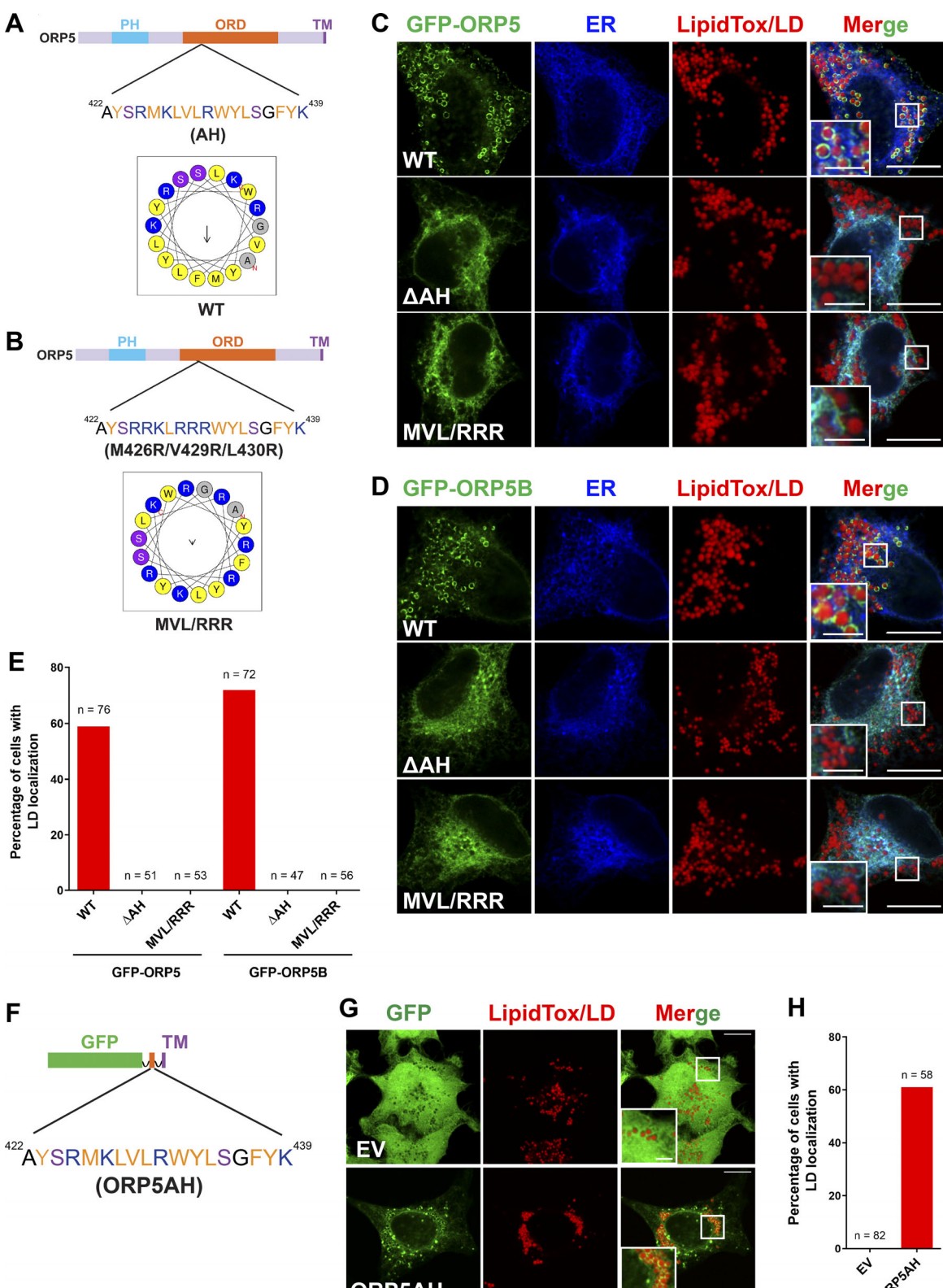

Figure 4. **The AH of ORP5-ORD and LD targeting of ORP5. (A)** Helical wheel representation of one of the amphipathic helices (aa 422–439) within ORP5-ORD generated at HeliQuest (http://heliquest.ipmc.cnrs.fr/). **(B)** Point mutations disrupting the amphipathic character of the helix within ORP5-ORD. **(C and D)** Confocal images showing the localizations of GFP-ORP5 or GFP-ORP5B with the AH deleted or disrupted in cells cotransfected with the ER marker DsRed-ER and treated with oleate for 16 h. LDs were stained with LipidTox DeepRed. Bars = 10 µm (insets, 2.5 µm). **(E)** Percentage of the cells expressing GFP-ORP5 or GFP-ORP5B with the AH deleted or disrupted that show LD localizations. **(F)** Construct of GFP-tagged ORP5AH fused with the transmembrane helix (TM) of ORP5. **(G)** Confocal images showing the localizations of GFP empty vector (EV) and GFP-ORP5AH shown in F in cells treated with oleate for 16 h. LDs were stained with LipidTox DeepRed. Bars = 10 µm (insets, 2.5 µm). **(H)** Percentage of cells expressing GFP EV or GFP-ORP5AH that show LD localizations. All data are representative of at least three independent experiments with similar results.

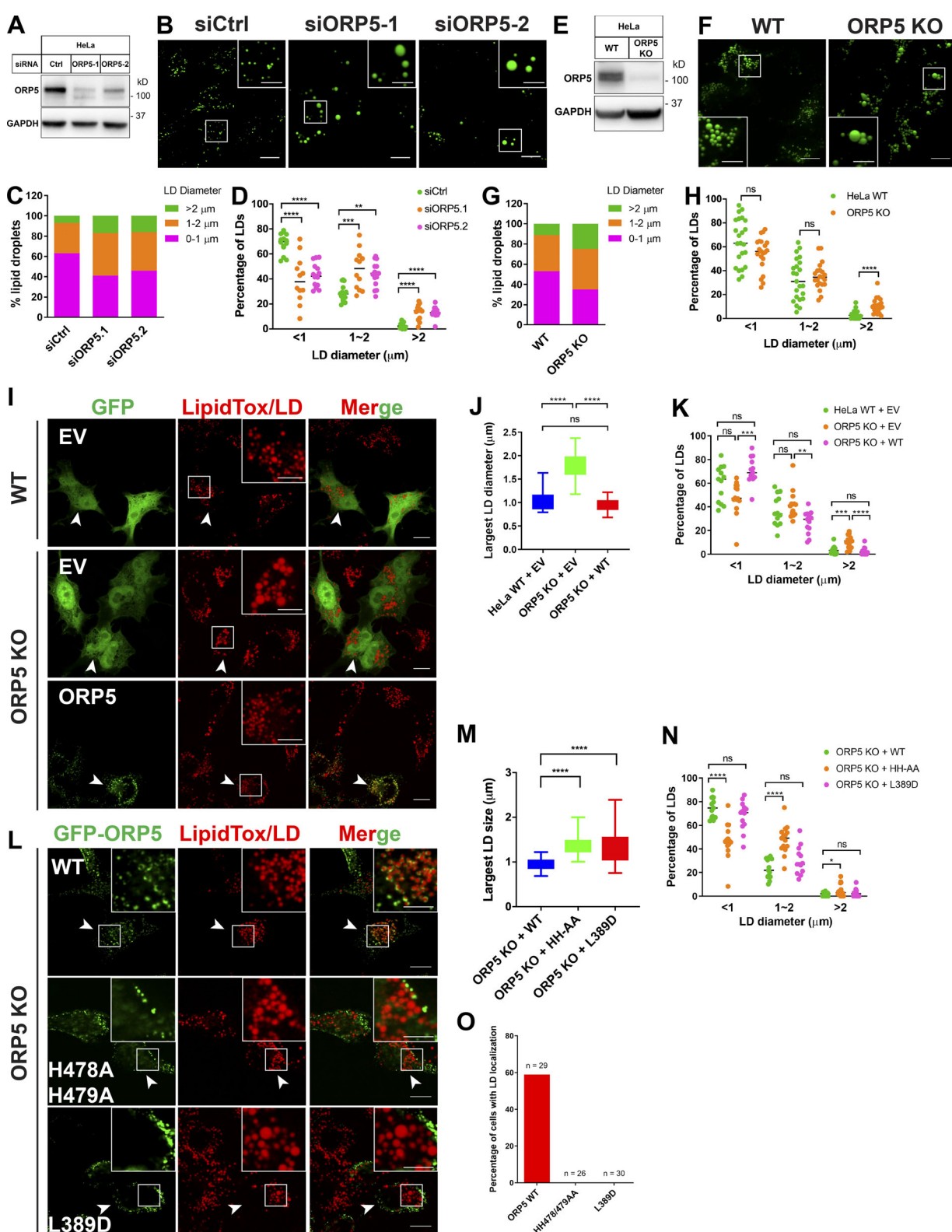

**Figure 5. ORP5 regulates LD size. (A)** Western blot analysis of ORP5 in cells treated with control or two different ORP5 siRNAs. **(B)** Confocal images of oleate-treated cells transfected with siRNAs as in A and stained with BODIPY fluorescent dye. Bars = 10 µm (insets, 5 µm). **(C)** Relative population of LD sizes from the results in B (~2,000 LDs from 10–20 cells). **(D)** Percentage of LDs with different diameters (<1, 1 to ~2, and >2 µm) in control and ORP5 knockdown cells shown in B. Mean ± SD. **, P < 0.01; ***, P < 0.001; ****, P < 0.0001. Each dot represents one cell (*n* = 13–25). **(E)** Western blot analysis of ORP5 in HeLa WT and ORP5 KO cells generated by the CRISPR/Cas9 system. **(F)** Confocal images of oleate-treated HeLa WT and ORP5 KO cells stained with BODIPY. Bars = 10 µm (insets, 5 µm). **(G)** Relative population of LD sizes from the results in E (~2,000 LDs from 10–20 cells). **(H)** Percentage of LDs with different diameters (<1, 1 to ~2, and >2 µm) in HeLa WT and ORP5 KO cells. Mean ± SD. ****, P < 0.0001. Each dot represents one cell (*n* = 21–23). **(I)**

Confocal images of oleate-treated HeLa WT and ORP5 KO cells expressing empty vector (EV) or GFP-ORP5. LDs were stained with LipidTox DeepRed. Bars = 10 µm (insets, 5 µm). **(J)** Sizes of the largest LDs in transfected cells in G. Mean ± SD. ****, P < 0.0001, n = 15–20. **(K)** Percentage of LDs with different diameters (<1, 1 to ~2, and >2 µm) in HeLa WT and ORP5 KO cells. Mean ± SD. **, P < 0.01; ***, P < 0.001; ****, P < 0.0001. Each dot represents one cell (n = 13–14). **(L)** Confocal images of oleate-treated ORP5KO cells expressing GFP-ORP5 or the mutants deficient for PI4P (H478A/H479A) or PS (L389D) transport. LDs were stained with LipidTox DeepRed. Bars = 10 µm (insets, 5 µm). **(M)** Sizes of the largest LDs in transfected cells in L. Mean ± SD. ****, P < 0.0001, n = 15–20. **(N)** Percentage of LDs with different diameters (<1, 1 to ~2, and >2 µm) in ORP5 KO cells expressing GFP-ORP5 WT or mutants. Mean ± SD. *, P < 0.05; ****, P < 0.0001. Each dot represents one cell (n = 13–15). **(O)** Percentage of transfected ORP5 KO cells expressing GFP-ORP5 WT or mutants that show LD localizations. All data are representative of at least three independent experiments with similar results.

GFP-P4M also accumulated around LDs in ORP5 KO cells but not in WT cells (Fig. 6, C and D). Unlike GFP-P4M, the majority of GFP-PLCPH concentrated on the PM where PI(4,5)P$_2$ is enriched, and very little of the probe was associated with LDs in either WT or ORP5 KO cells (Fig. 6, C and D). Given the known role of ORP5 in PS transfer, we examined intracellular localization of PS using the marker GFP-evt-2PH (Uchida et al., 2011) in cells treated with oleate. Interestingly, a relatively strong association of GFP-evt-2PH with LDs was found in WT cells, but not in ORP5 KO cells, opposite to GFP-P4M (Fig. 6, E and F). Overall, the results of these studies indicate that ORP5 can regulate the abundance of PI(4)P and PS on LDs.

### PI4K2A regulates the LD-associated pool of PI(4)P

The detection of PI(4)P on LDs in ORP5-deficient cells prompted us to understand how this pool of PI(4)P is generated. PI(4)P in mammals is produced by four PI4K isoforms: PI4K2A, PI4K2B, PI4KA/PI4KIIIα, and PI4KB/PI4KIIIβ; and the majority of PI(4)P in humans is produced by PI4K2A (Boura and Nencka, 2015). PI4K2A was previously identified on LDs isolated from human fatty liver tissue (Su et al., 2014), so we coexpressed PI4K2A and a PI(4)P probe, SidC-GFP (Luo et al., 2015), in HeLa cells and induced LD formation by oleate treatment. More than 60% of cells expressing the WT kinase had LDs surrounded by the PI(4)P probe (Fig. 7, A and B). This phenomenon was not seen in cells expressing the kinase-dead mutant PI4K2A(K152A) (Fig. 7, A and B). Airyscan microscopy revealed that GFP-PI4K2A mainly localized to the ER in WT cells, but it became more punctate in ORP5-deficient cells (Fig. S5 G). In oleate-treated ORP5-deficient cells, these GFP-PI4K2A puncta were found very close to LDs (Fig. 7 C). We then silenced PI4K2A expression by siRNA and examined the distribution of PI(4)P in ORP5 KO cells (Fig. 7, D–F). The association of GFP-P4M with LDs became markedly weaker in ORP5 KO cells when PI4K2A was depleted by siRNAs (Fig. 7, F and G). Moreover, PI4K2A depletion noticeably increased the size of LDs in WT cells, but not in ORP5 KO cells where the LDs were already enlarged (Fig. 7, H and I). Finally, depletion of the other three PI4 kinases did not impact LD morphology in WT cells (not depicted). Together, these results suggest that PI4K2A is responsible for the pool of PI(4)P associated with LDs.

## Discussion

Among all cellular organelles, LDs are highly unique in that each LD is bounded by a monolayer of phospholipids, which play key roles in LD budding, growth, degradation, and protein targeting (Bersuker et al., 2018; Gao et al., 2019; Walther et al., 2017).

However, how the phospholipid composition of LDs is established and maintained is largely unknown. While certain steps of phospholipid synthesis can take place on LDs, most phospholipids are synthesized in the ER. Therefore, there must be a dynamic lipid exchange between the ER and LDs. Here, we identify ORP5 as the only OSBP/ORP protein that is highly concentrated at ER–LD contacts upon oleate loading. Moreover, our results demonstrate the existence of phosphoinositides on LDs for the first time and suggest that ORP5 regulates LD growth by mediating the countertransport of PS and PI(4)P between the ER and LDs.

The phospholipid monolayer of LDs is decorated with proteins that influence LD dynamics as well as lipid homeostasis, and many of these proteins are associated with metabolic disorders (Xu et al., 2018). Although the pathways of proteins targeting the LD surface are being recognized, how the targeting specificity is determined has yet to be fully elucidated (Kory et al., 2016). We provide several lines of evidence demonstrating that ORP5 specifically localizes to the ER–LD contact sites. As shown by confocal microscopy and immunofluorescence, both overexpressed and endogenous ORP5 localizes to LDs in oleate-treated cells. APEX EM data also clearly reveal that GFP-ORP5 is enriched on the LD surface, particularly in the regions connecting the ER and LDs. Notably, among all 12 members of the OSBP/ORP family, only ORP5 is highly enriched at ER–LD contacts. ORP8, a structurally similar and closely related family member of ORP5 (Pietrangelo and Ridgway, 2018; Yan et al., 2008), failed to target to LDs. We further identify an AH domain within the ORD region of ORP5 that appears to be necessary and sufficient for LD targeting (Barneda et al., 2015; Hinson and Cresswell, 2009; Prevost et al., 2018; Rowe et al., 2016). However, it is important to note that this AH domain may also be important for stabilizing the ORP5-ORD based on modeling with the ORD of Osh6p (not depicted). Future structural and functional analyses of ORP5-ORD are needed to determine the precise mechanism by which ORP5 is targeted to the LDs. Finally, given that ORP5 is a tail-anchored protein, overexpression of ORP5 might have saturated ORP5 targeting sites on the ER, causing it to localize to the LDs instead. However, this is a remote possibility because ORP8 has an almost identical tail to ORP5 but has never been detected on LDs.

ORP5 is an LTP that operates at contact sites between the ER and other organelles including the PM, late endosomes, and mitochondria (Chung et al., 2015; Du et al., 2011; Galmes et al., 2016; Ghai et al., 2017; Sohn et al., 2018). Specifically, ORP5 has been recently shown as a PS/phosphoinositides exchanger between the ER and the PM (Chung et al., 2015; Ghai et al., 2017; Sohn et al., 2018). This activity of ORP5 is important in

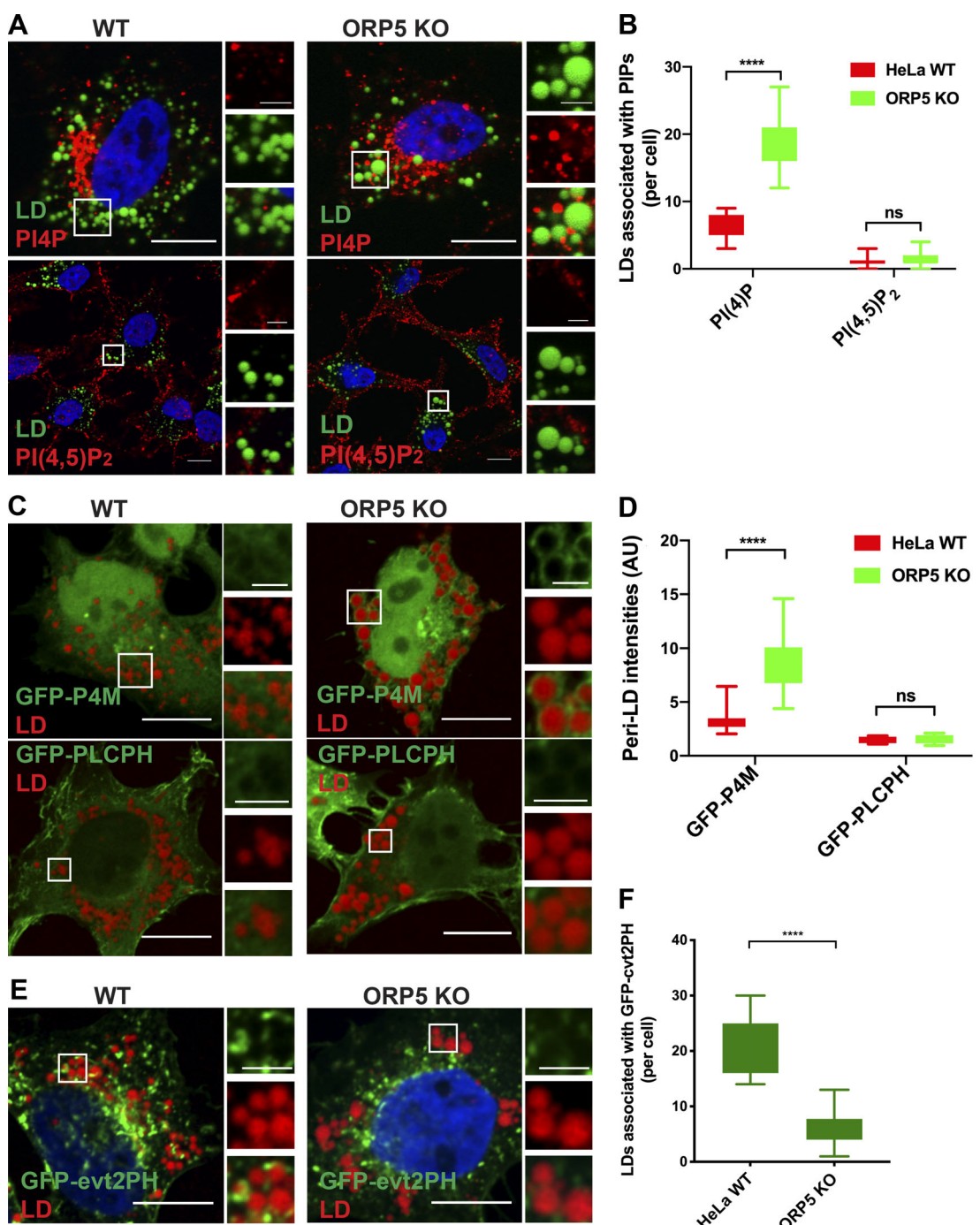

Figure 6. **ORP5 controls the amount of PI(4)P and PS on LDs. (A)** Immunofluorescence of PI(4)P and PI(4,5)P$_2$ in oleate-treated HeLa WT and ORP5 KO cells. LDs were stained with BODIPY. Bars = 10 μm (inlay, 2.5 μm). **(B)** Quantitation of LDs per cell associated with PI(4)P or PI(4,5)P$_2$ puncta. Mean ± SD. ****, P < 0.0001, n = 14–15. **(C)** Confocal images of oleate-treated HeLa WT and ORP5 KO cells expressing the PI(4)P sensor GFP-P4M and the PI(4,5)P$_2$ sensor GFP-PLCPH. LDs were stained with LipidTox DeepRed. Bars = 10 μm (inlay, 2.5 μm). **(D)** Quantitation of GFP-P4M or GFP-PLCPH intensities on the surface of LDs in C. ****, P < 0.0001, n = 37–38. **(E)** Confocal images of oleate treated WT HeLa and ORP5KO cells expressing the PS sensor GFP-evt2PH. LDs were stained with LipidTox DeepRed. Bars = 10 μm (inlay, 2.5 μm). Mean ± SD. **(F)** Quantitation of LDs associated with the PS sensor in G. Mean ± SD. ****, P < 0.0001, n = 15–20. All data are representative of at least three independent experiments with similar results.

maintaining the levels of phosphoinositides such as PI4P and/or PI(4,5)P$_2$ on the PM (Ghai et al., 2017; Sohn et al., 2018). Our results here suggest that ORP5 functions at ER–LD contact sites to deliver PS to LD surface and to bring back PI(4)P to the ER: LD-associated PI(4)P was increased whereas PS was decreased in ORP5-deficient cells. This notion is further supported by findings that the ORP5 mutants defective for PI(4)P or PS transport could not restore normal LD morphology in ORP5-deficient cells. Although a clear difference in LD-associated PI(4)P and PS was detected by antibodies and/or sensors between normal and

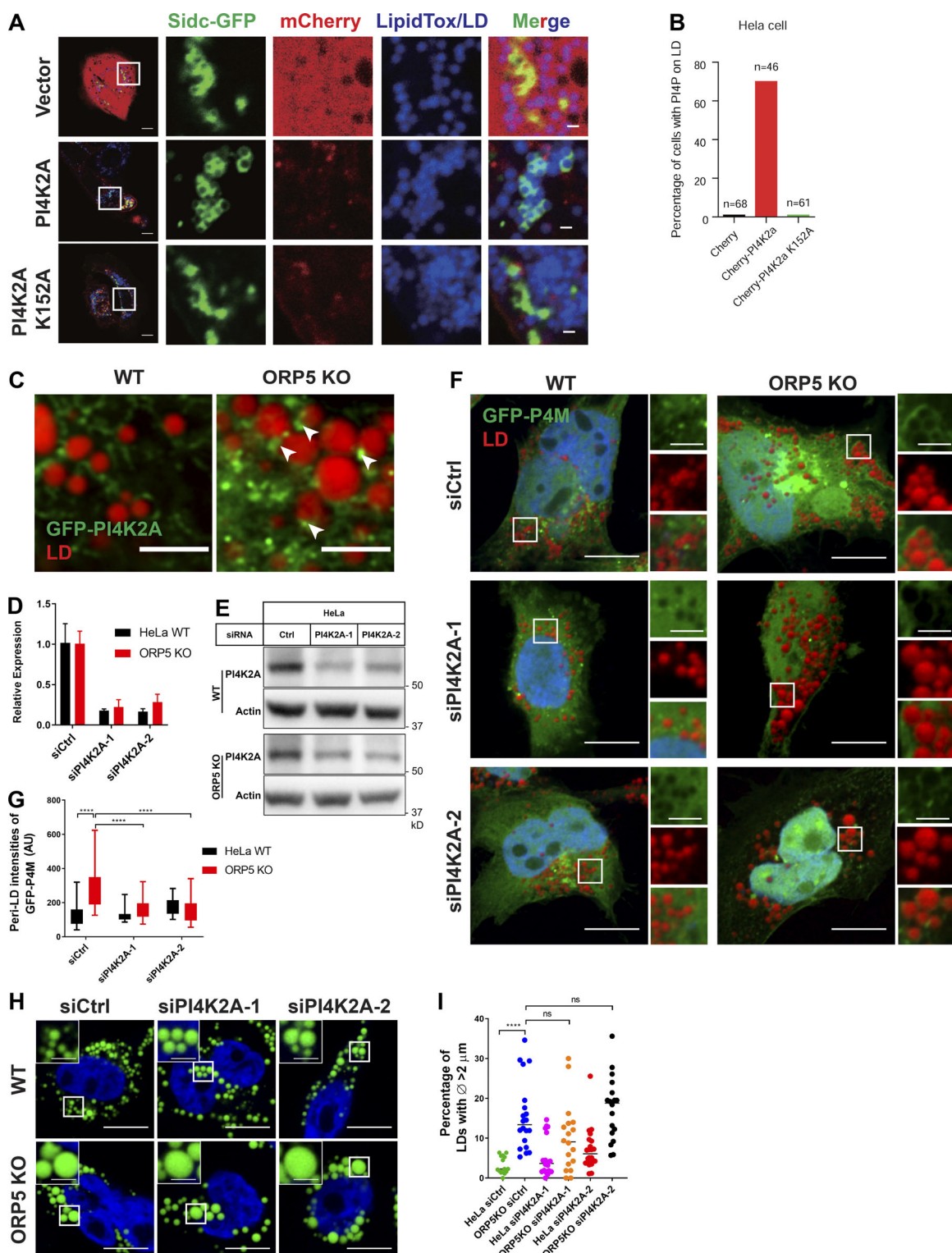

Figure 7. **PI4K2A controls PI(4)P association with LDs. (A)** Overexpression of mCherry-PI4K2A or -PI4K2A(K152A) together with the PI(4)P probe Sidc-GFP in oleate-treated cells. LDs were stained with LipidTox DeepRed. Bars = 10 µm (inlay, 2 µm). **(B)** Percentage of transfected cells with PI(4)P on LDs in A. **(C)** Airyscan confocal images of HeLa WT and ORP5 KO cells expressing GFP-PI4K2A and treated with oleate for 16 h. LDs were stained with LipidTox DeepRed. Arrowheads indicate LD-associated PI4K2A. Bars = 10 µm. **(D)** RT-PCR analysis of *PI4K2A* expression in HeLa and ORP5 KO cells treated with PI4K2A siRNAs for 48 h (*n* = 3). Mean ± SD. **(E)** Western blot analysis of HeLa WT and ORP5 KO cells treated with control or PI4K2A siRNAs for 48 h. **(F)** Confocal images of oleate-treated HeLa and ORP5KO cells expressing the PI4P sensor GFP-P4M. Cells were treated with siRNAs for 72 h. LDs were stained with LipidTox DeepRed. Bars = 10 µm (inlay, 2.5 µm). **(G)** Quantitation of GFP-P4M intensities surrounding LDs in F. Mean ± SD. ****, P < 0.0001, *n* > 51–52. **(H)** Confocal images of oleate-treated HeLa and ORP5KO cells transfected with siRNAs for 72 h. LDs were stained with BODIPY. Bars = 10 µm (inlay, 2.5 µm). **(I)** Percentage of LDs with diameters >2 µm in oleate-treated HeLa and ORP5KO cells transfected with siRNAs for 72 h. Mean ± SD. All data are representative of at least three independent experiments with similar results.

ORP5-deficient cells, it should be noted that these sensors/antibodies as well as detection methods have their limitations. Our attempt to quantify lipids of isolated LDs by lipidomics failed to detect any consistent and significant difference in PS or PI(4)P levels between WT and ORP5-deficient cells (not depicted). This could be due to the contamination of isolated LDs by ER membranes or to the technical limitations to distinguish subtle changes of PS and PI(4)P. Moreover, the abundance of PS and PI(4)P on the LD surface is very low (<1%; Bartz et al., 2007). Therefore, future studies with better detection techniques are required to further examine the level and distribution of PI(4)P and PS on the LD surface. Finally, while ORP5 has also been proposed to directly transfer PI(4,5)P$_2$, we could not detect PI(4,5)P$_2$ or other phosphoinositides on LDs (data not shown) in WT or ORP5-deficient cells. It is likely that PI(4)P is the only abundant phosphoinositide present on the LD surface, or that the techniques we used are not sensitive enough to detect LD surface PI(4,5)P$_2$.

Why is ORP5, and to a much lesser extent, ORP2, but not other ORPs, required at ER–LD contact sites upon oleate loading? Although the crystal structure of ORP5 is not available and the exact ligand of ORP5-ORD remains to be determined, current evidence suggests that ORP5 transports PS. Interestingly, however, the level of PS on the LD surface is very low (<1% of total phospholipids; Bartz et al., 2007). Thus, PS may play an important signaling/regulatory role rather than a structural role on LDs. Given its negative charge, intracellular PS can regulate the localization and activity of a number of proteins including AKT, PKC, and phospholipases (Huang et al., 2011; Leventis and Grinstein, 2010). Notably, PS is also known to form microdomains with cholesterol and regulates its distribution (Maekawa and Fairn, 2015). However, we were not able to detect changes in cholesterol on LDs in ORP5-deficient cells (Fig. S5 H). While the exact role of PS on LDs remains to be examined, it is clear that ORP5-mediated PS delivery to LDs is needed for the proper function of LDs.

Besides ORP5, a few other LTPs have been located at the ER–LD contact sites. ORP2, a cytosolic sterol carrier known to deliver cholesterol from endolysosomal compartments to the PM in exchange for PI(4,5)P$_2$ (Wang et al., 2019), has been shown to localize at LD surface/ER–LD contact sites (Hynynen et al., 2009; Koponen et al., 2019). However, the association of ORP2 with LDs was much less prominent than that of ORP5 in our hands. Other interesting proteins include VPS13A and VPS13C, which have been recently identified as novel LTPs localizing at ER–LD contact sites (Kumar et al., 2018). By binding to the ER and LD surface through their N- and C-terminal regions, respectively, these two proteins may mediate the bulk transfer of glycerophospholipids between the two organelles (Kumar et al., 2018). ATG2 is another putative LTP associated with the LDs (Velikkakath et al., 2012). It is likely that these proteins deliver different lipids, e.g., phosphatidylcholine, phosphatidylethanolamine, and cholesterol, to LDs. Further work is needed to investigate the spatial and functional relationships between ORP5, ORP2, VPS13s, and ATG2.

Phosphoinositides are key signposts of cellular organelles (Balla, 2013). The existence of phosphoinositides on LDs have

been implicated but never demonstrated (Ren et al., 2014). Here, we detected PI(4)P on the LD surface in ORP5-deficient cells with an antibody and with a well-established PI(4)P sensor. Moreover, knocking down PI4K2A reduced PI(4)P association with LDs in ORP5-deficient cells. Finally, PI4K2A overexpression promoted PI(4)P association with the LDs. These data suggest that PI(4)P produced by PI4K2A does exist on LDs in WT cells but is rapidly consumed/removed by ORP5. Thus, our data identify a molecular pathway by which the level of PI(4)P on the LD surface is regulated. The presence of PI(4)P on the LD surface may serve as a signal to recruit proteins to the LDs, thereby contributing to the identity of LDs (Penno et al., 2013). However, given that LD-associated PI(4)P is readily detectable only under ORP5 deficiency, this pool of PI(4)P may be primarily used by ORP5 to drive the transport of PS. The dramatic effect of ORP5 on the LD pool of PI(4)P is consistent with the new theme that the OSBP/ORP family are major regulators of cellular phosphoinositides. For instance, OSBP consumes about half of the total cellular pool of PI4P (Mesmin et al., 2017), and ORP2 may control ~40% of cellular PI(4,5)P$_2$ (Wang et al., 2019).

In summary, the data presented here show that ORP5, an ER-anchored LTP, localizes to ER–LD contacts to deliver PS to LDs at the expense of PI(4)P. Most importantly, our results demonstrate for the first time that PI(4)P, produced by PI4K2A, does exist on LDs.

## Materials and methods
### Cell culture and transfection
All cell lines were originally obtained from the American Type Culture Collection except the HEK-293 line, from Life Technologies. Monolayers of cells were maintained in DMEM supplemented with 10% FBS, 100 units/ml penicillin, and 100 μg/ml streptomycin sulfate in 5% CO$_2$ at 37°C. DNA transfection was performed using Lipofectamine LTX and Plus Reagent (Life Technologies) according to the manufacturer's instruction. siRNA transfection was performed in cells grown in full serum medium according to standard methods using Lipofectamine RNAiMAX transfection reagent (Life Technologies).

### RNAi and cDNA constructs
MISSION siRNA Universal Negative Control and siRNAs against ORP5 (targeting sequences: SASI_Hs02_00365256, 5′-TGG GTGGGAAGGTCACCAT-3′; SASI_Hs01_00039221, 5′-CGC CCACTGCAAAGGAATC-3′) and PI4K2A (targeting sequences: SASI_Hs01_0 0190417, 5′-CAATGACAACTGGCT GATT-3′; SASI_Hs01_00190418, 5′-GCTACAAAGATGCAG ACTA-3′) were obtained from Sigma-Aldrich. GFP- or mCherry-tagged ORP1L-ORP11 and PI4K2A were constructed by subcloning the respective coding cDNA into pEGFP-C1 or pmCherry-C1 vectors (Clontech). Point mutations or domain deletions of ORP5 (L389D, H478A/H479A, ΔPH, ΔORD, ΔPHΔORD, ΔAH, and MVL/RRR), ORP5B (L321D, H410A/H411A, ΔAH, and MVL/RRR), ORP8 (ORP5AH), GFP-ORP5AH-TM, and PI4K2A (K152A) were generated by site-directed mutagenesis/deletions/insertions. Details of other constructs were described previously, including MAPPER, a GFP-conjugated marker for

ER–PM contact sites (Chang et al., 2013), GFP-Sec61β (GFP-fused tail-anchored protein Sec61β; Ma and Mayr, 2018), APEX2-GBP (a fusion of APEX2 with a GBP; Ariotti et al., 2015), GFP-ADRP (GFP-tagged adipose differentiation related protein; Gong et al., 2011), GFP-Hpos (GFP-tagged hydrophobic domain of associated with lipid droplet protein 1, which is fused with the last 20 residues of caveolin-1; Kassan et al., 2013), GFP-P4M (GFP-conjugated P4M domain consisting of residues 546–647 of *Legionella pneumophila* SidM; Hammond et al., 2014), GFP-PLCPH (GFP-tagged PH domain and phospholipase C δ, which is known to specifically bind PtdIns(4,5)P2; Stauffer et al., 1998), GFP-evt2PH (GFP-tagged tandem fusion of evectin-2 [2 × PH]; Uchida et al., 2011), DsRed-ER (a fusion of the ER targeting sequence of calreticulin fused to the 5′ end of DsRed2 and the ER retention sequence, KDEL, fused to the 3′ end of DsRed2; Du et al., 2011), and Sidc-GFP (PI(4)P-binding protein, SidC, tagged with GFP; Luo et al., 2015).

## Immunofluorescence
For immunostaining of ORP5, cells were transfected with control or ORP5 specific siRNAs for 48 h. After an overnight oleate treatment, cells were washed with PBS and fixed for 15 min with 4% PFA in PBS and permeabilized for 10 min with 0.1% Triton X-100 in PBS. Cells were washed with PBS and blocked with 4% FBS in PBS for 1 h at RT, followed by overnight incubation with the primary antibody (Sigma-Aldrich, HPA038335, 4 µg/ml) diluted in the blocking solution. Cells were washed with PBS (3× 10 min) and incubated for 1 h at RT with Alexa Fluor 568 secondary antibody (Invitrogen). Cells were washed with PBS (3 × 10) min and stained with BODIPY for LDs (see below), followed by mounting in ProLong Gold antifade reagent with or without DAPI (Invitrogen).

For immunostaining of PI(4)P and PI(4,5)P$_2$, all steps were performed at RT. Cells grown on coverslips in 1 ml medium were fixed by the addition of 1 ml of 4% PFA in PBS for 15 min. Cells were washed three times with PBS containing 50 mM NH$_4$Cl, followed by permeabilization for 5 min with 20 µM digitonin in PBS. After three rinses with PBS to remove digitonin, cells were blocked for 45 min in PBS containing 5% normal goat serum (NGS) and 50 mM NH$_4$Cl. Primary anti-PI(4)P antibody (Echelon Biosciences, Z-P004, 1:50) and anti-PI(4,5)P$_2$ (Echelon Biosciences, Z-P045, 1:100) were diluted in PBS containing 5% NGS and applied to cells for 1 h. After three washes with PBS, incubation with Alexa Fluor 568 secondary antibody (Invitrogen) diluted in PBS containing 5% NGS was performed for 45 min. Cells were then washed three times with PBS and postfixed in 2% PFA in PBS for 10 min at RT, followed by three washes with PBS containing 50 mM NH$_4$Cl and one rinse in distilled water. Cells were mounted in ProLong Gold antifade reagent.

## LD staining
Unless otherwise stated, cells grown on coverslips were treated with 300 µM oleate overnight (16–18 h), followed by fixation with 4% PFA diluted in PBS for 15 min at RT. Cells were washed with PBS three times and permeabilized with 0.2% Triton X-100 diluted in PBS for 10 min. LDs were stained with either 1 µg/ml of BODIPY 493/503 (Invitrogen) in PBS for 10 min

at RT or HCS LipidTOX Deep Red Neutral Lipid Stain (Invitrogen; 1:500–1,000) diluted in PBS for 30–60 min. Cells were mounted in ProLong Gold antifade reagent with or without DAPI.

## Filipin staining
In the experiments involving free cholesterol staining, fixed cells were stained with freshly prepared ∼50 µg/ml of filipin (a fluorescent dye that specifically binds to cellular free cholesterol) in PBS for 1 h at RT, followed by LD staining by BODIPY 493/503 and mounting with ProLong Gold antifade reagent.

## Microscopy and image analysis
Confocal microscopy was performed using an FV1200 laser scanning confocal microscope (Olympus) or an LSM 780 upright confocal laser scanning microscope (Zeiss). A 100× or 63×/1.4 oil-immersion objective was used for all imaging at RT. The fluorochromes used were Alexa Fluor 488 (BODIPY 493/503), Alexa Fluor 568, Cy5 (LipidTox DeepRed), EGFP, Texas Red (mCherry and DsRed), and DAPI (filipin). For comparisons of fluorescence intensities between different samples, images were collected during a single session at identical excitation and detection settings. GFP-P4M intensities surrounding LDs were measured using Fiji software. Briefly, two circles were placed onto an individual LD, with the bigger one covering the LD and GFP signals on the LD surface and the smaller one covering the LD only. The intensity of GFP-P4M on the LD surface was determined by normalizing the intensity difference of the two circles with the LD diameter. FRAP experiments were performed with the LSM 780 on a 100×/1.4 oil-immersion objective at 37°C with 5% CO$_2$. Cells were imaged in DMEM/10% FBS supplemented with LipidTox DeepRed (1:500–1,000 dilution). Zeiss Zen Black software was used to analyze the recovery of GFP-ORP5 signal on the LD surface. Nonbleached regions in the same cells were included in the analysis to determine the photobleaching level in each time series. After normalization, GFP-ORP5 intensity was plotted against time. The experiment was repeated more than three times, and at least three cells were measured in each experiment. For Airyscan high-resolution confocal microscopy, an LSM 800 confocal microscope system equipped with a 63×/1.4 oil objective (Zeiss) was used at RT. 3D reconstruction of LDs associated with GFP-ORP5 from Airyscan confocal stack images was performed using Imaris (surface rendering, v9.1). For all other microscopic experiments, image acquisition and compiling were performed using Fiji and Adobe Illustrator software unless otherwise stated.

## Generation of ORP5 KO cells
HeLa ORP5 KO cells were generated by the CRISPR/Cas9 system. Briefly, sgRNA sequences were designed by analyzing exon sequences close to the start codons of ORP5 (NCBI Reference Sequence: NM_020896) using https://www.synthego.com/products/bioinformatics/crispr-design-tool. DNA oligonucleotide sequence 5′-TCAGAAAGTCGACCCCCGGA-3′ was chosen as sgRNA to knock out ORP5. Corresponding DNA oligonucleotides were synthesized with BbsI sites flanking the forward and reverse oligonucleotides. The synthesized DNA oligonucleotides were annealed and subcloned into the sgRNA expression vector pSpCas9(BB)-2A-GFP

(PX458), which was a gift from Feng Zhang, McGovern Institute for Brain Research, Massachusetts Institute of Technology, Cambridge, MA (Addgene plasmid 48138; Ran et al., 2013). PX458-sgRNA was verified by sequencing and transfected into WT HeLa cells. 48 h after transfection, FACS of HeLa cells was performed with a BD Influx Cell Sorter (BD Biosciences), and a single cell with GFP signal was sorted into each well of 96-well plates. Individual cell clones were expanded and screened for ORP5 deficiency by Western blotting analysis. Three positive ORP5 KO clones were used in this study.

### LD purification
The purification of LDs from cells treated with oleate overnight was performed using the Lipid Droplet Isolation Kit (Cell Biolabs, MET-5011) according to the manufacturer's instructions. The volumes of LD fractions from different samples were normalized against protein contents determined by the bicinchoninic acid assay.

### Immunoblot analysis and antibodies
Samples were mixed with 2× Laemmli buffer and subjected to 7.5% or 10% SDS-PAGE. After electrophoresis, the proteins were transferred to Hybond-C nitrocellulose filters (GE Healthcare). Incubations with primary antibodies were performed at 4°C overnight. Primary antibodies used were rabbit polyclonal to ORP5 (Sigma-Aldrich, HPA038335), GAPDH (Cell Signaling Technology, 2118), Calnexin (Cell Signaling Technology, 2433), β-actin (Cell Signaling Technology, 4970), and CGI-58/ABHD5 (Proteintech, 12201-1-AP) and mouse monoclonal to GFP (Santa Cruz Biotechnology, sc-9996). Secondary antibodies were peroxidase-conjugated AffiniPure donkey anti-rabbit or donkey anti-mouse IgG (H+L; Jackson ImmunoResearch Laboratories) used at a 1:5,000 dilution. The bound antibodies were detected by enhance chemiluminescence Western blotting detection reagent (GE Healthcare or Merck Millipore) and visualized with Molecular Imager ChemiDoc XRS+ (Bio-Rad Laboratories).

### APEX EM
APEX EM was performed as previously described (Ariotti et al., 2015). Briefly, cells were fixed in 2.5% glutaraldehyde and washed three times in PBS and once in 1 mg/ml DAB solution in PBS for 5 min at RT. Cells were then incubated with 1 mg/ml DAB in PBS in the presence of 0.02% (vol/vol) $H_2O_2$ for 30 min at RT, followed by washing in PBS and postfixation in 1% osmium tetroxide for 2 min. Cells were washed again, followed by serial dehydration in increasing percentages of ethanol, serial infiltration with LX112 resin in a Pelco Biowave microwave, and polymerization overnight at 60°C. Ultrathin sections were cut on an ultramicrotome (EM UC6, Leica Microsystems) and imaged using a JEOL1011 electron microscope (JEOL) at 80 kV.

### TAG measurement
TAG measurement was performed by TLC and a fluorometric neutral lipid quantification kit (Cell Biolabs, #STA-617). Briefly, the monolayer of cells grown in 60-mm dishes was treated with oleate overnight. Cells were washed twice with buffer A (0.15 M NaCl, 0.05 M Tris-HCl, and 2 mg/ml BSA, pH 7.4) and once with

buffer B (0.15 M NaCl and 0.05 M Tris-HCl, pH 7.4). The cells were then incubated with 2 ml of hexane-isopropanol (3:2) at RT for 30 min. The organic solvent was collected, and cells were incubated with another 1 ml of the same solvent for 15 min at RT. The two organic solvent extracts were combined in 2-ml glass vials, which were kept in the fume hood until the solvent was evaporated to dryness. The cells remaining in each well were harvested using 1 ml of 0.1 M NaOH, and aliquots were removed for protein determination by the bicinchoninic acid protein assay. For TLC measurement, the lipids in each vial were resuspended in 60 µl of hexane and normalized to the total amount of proteins. The lipids and standards were spotted on a Silica Gel 60 F254 TLC plate. The plate was developed in heptane:diethyl ether:acetic acid (90:30:1) and then stained with iodine vapor in a glass chamber for 10–20 min. The stained plate was scanned, and the TAG bands corresponding to the standard were analyzed by densitometry using Fiji software. For fluorometric assay, the lipids in each vial were resuspended in 100 µl methanol/chloroform mixture (2:1) and the TAG content was assayed according to the manufacturer's protocol. The values were normalized to total cellular proteins.

### Statistical analysis
Statistical analysis between groups was performed using Prism 6 for Windows v8.0 (GraphPad Software) with Student's unpaired $t$ tests or one-way ANOVA. Data are expressed as mean + SD unless otherwise stated.

### Online supplemental material
Fig. S1 shows LD localization screen of mCherry-tagged ORPs in HeLa cells. Fig. S2 shows LD targeting of GFP-ORP5 in different mammalian cell lines. Fig. S3 shows the characterization of LD targeting of ORP5 and the role of the AH. Fig. S4 shows the role of ORP5AH in LD size regulation and the effect of ORP5 deficiency on early LD formation. Fig. S5 shows the effect of ORP5 deficiency on TAG synthesis; the localizations of DGAT1, DGAT2, and PI4K2A; and cholesterol distribution.

## Acknowledgments
We thank the Biomedical Imaging Facility at the University of New South Wales Mark Wainwright Analytical Centre, Australia, the Center of Biomedical Analysis at Tsinghua University, China, and the Centre for Microscopy and Microanalysis at the University of Queensland, Australia, for scientific and technical assistance.

H. Yang is supported by project grants from the National Health and Medical Research Council of Australia (APP1041301, 1141939, and 1144726) and by a National Health and Medical Research Council Senior Research Fellowship (1058237). P. Li is supported by grants from the National Natural Science Foundation of China (31690103, 31430040, and 31621063) and the National Key R&D Program of China (2016YFA0502002). R.G. Parton is supported by grants and a fellowship from the National Health and Medical Research Council of Australia (APP1037320, APP1058565, and APP569542) and by the Australian Research Council Center of Excellence in Convergent Bio-Nano Science and Technology.

The authors declare no competing financial interests.

Author contributions: X. Du, L. Zhou, Y.C. Aw, J. Rae, W. Wang, S.E. Hancock, A. Zadoorian, Y. Xu, X. Chen, H.Y. Mak, and B. Osborne performed experiments. X. Du, R.G. Parton, N. Turner, J.W. Wu, P. Li, and H. Yang designed research and analyzed data. X. Du and H. Yang wrote the manuscript together with all other authors.

Submitted: 22 May 2019

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

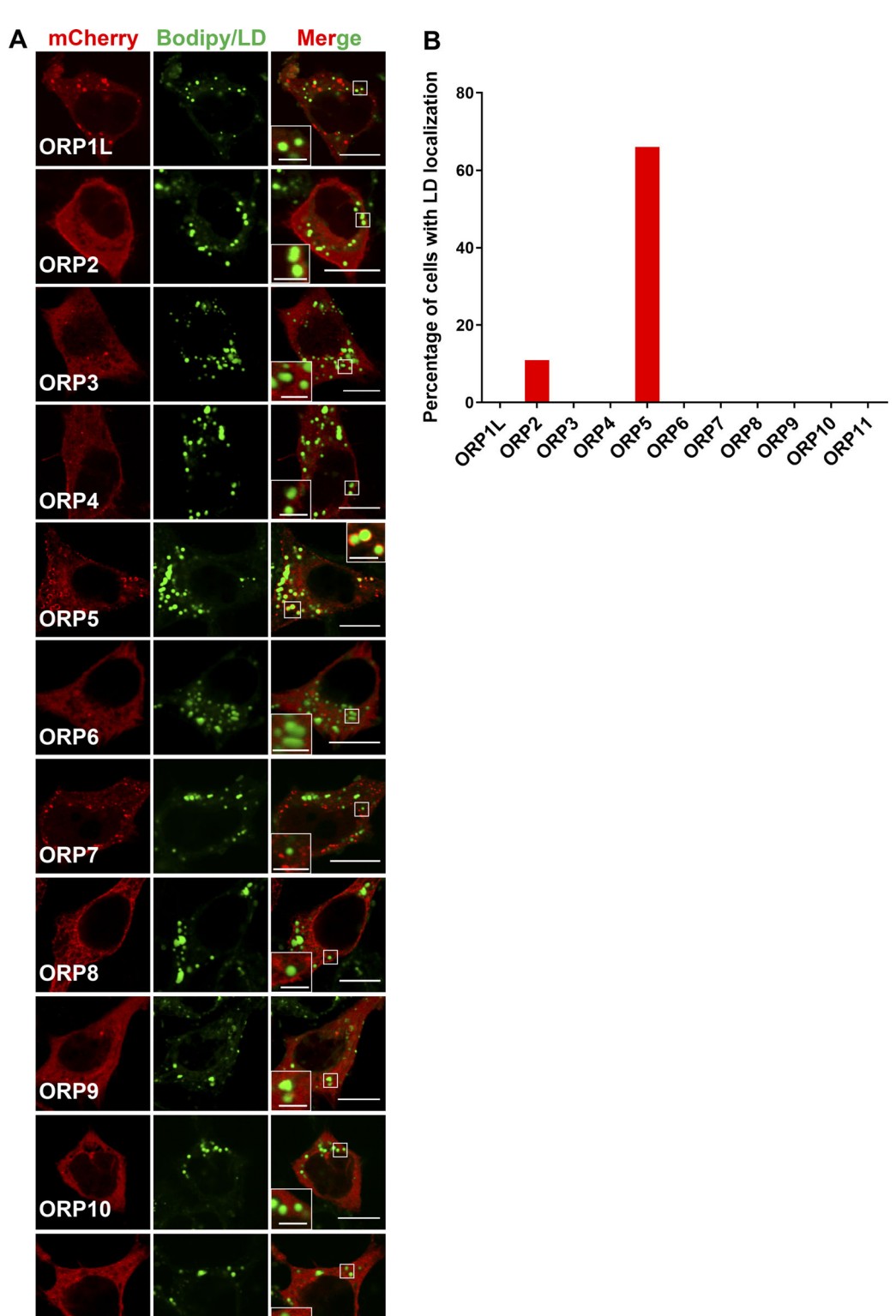

Figure S1.  **Localization of mCherry-ORPs in HeLa cells. (A)** HeLa cells were transfected with mCherry-tagged ORPs (ORP1L-ORP11) for 24 h, then treated with oleate for 16 h. Bars = 10 µm (insets, 2.5 µm). **(B)** Percentage of mCherry- or GFP-ORPs transfected cells that show LD localizations ($n$ = 10–27).

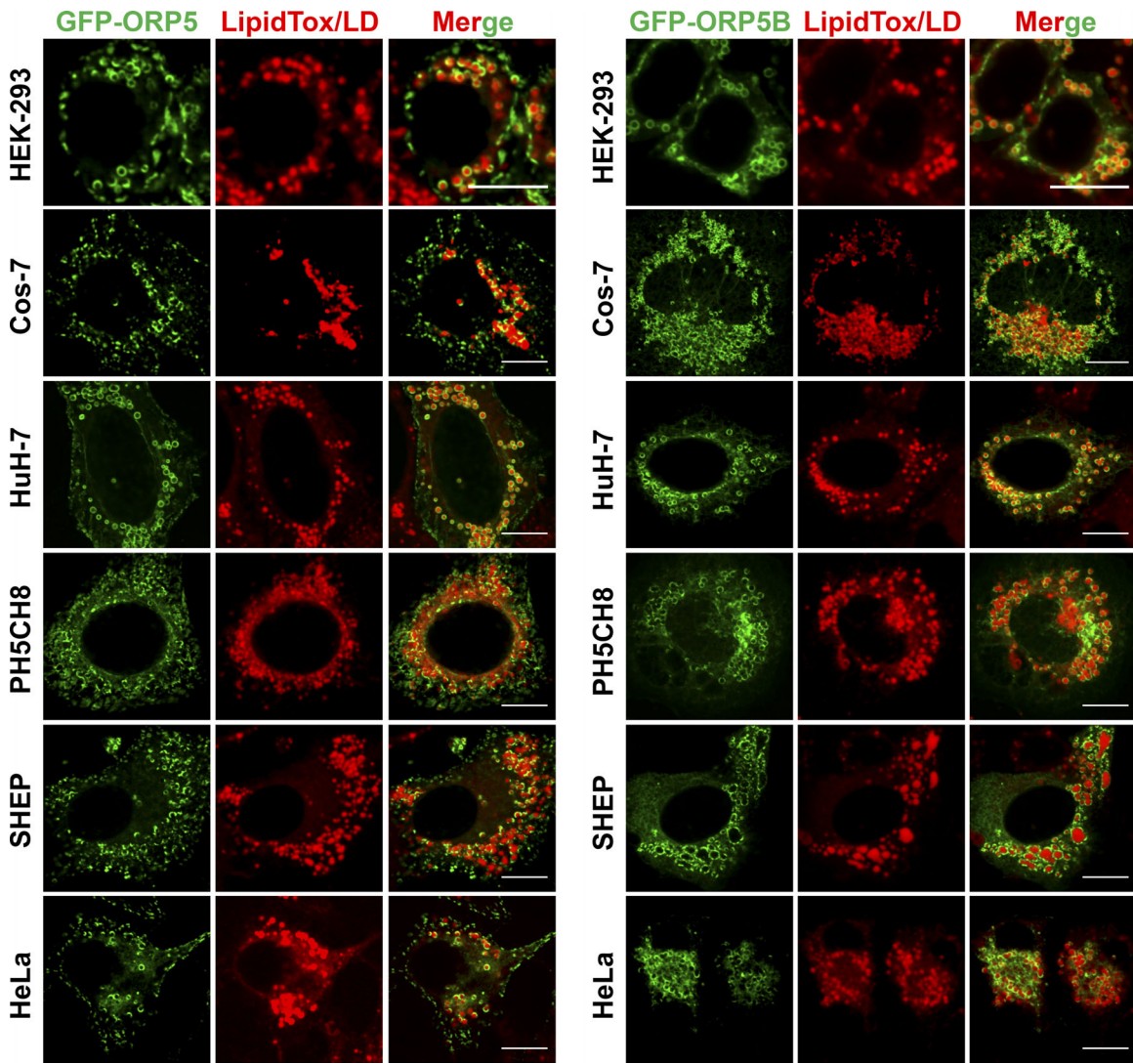

Figure S2.   **GFP-ORP5 targets to LDs in different mammalian cell lines.** HEK-293, Cos-7, Huh7, PH5CH8, SHEP, and HeLa cells were transfected with GFP-ORP5 or GFP-ORPB for 24 h, then treated with oleate for 16 h. Bars = 10 μm.

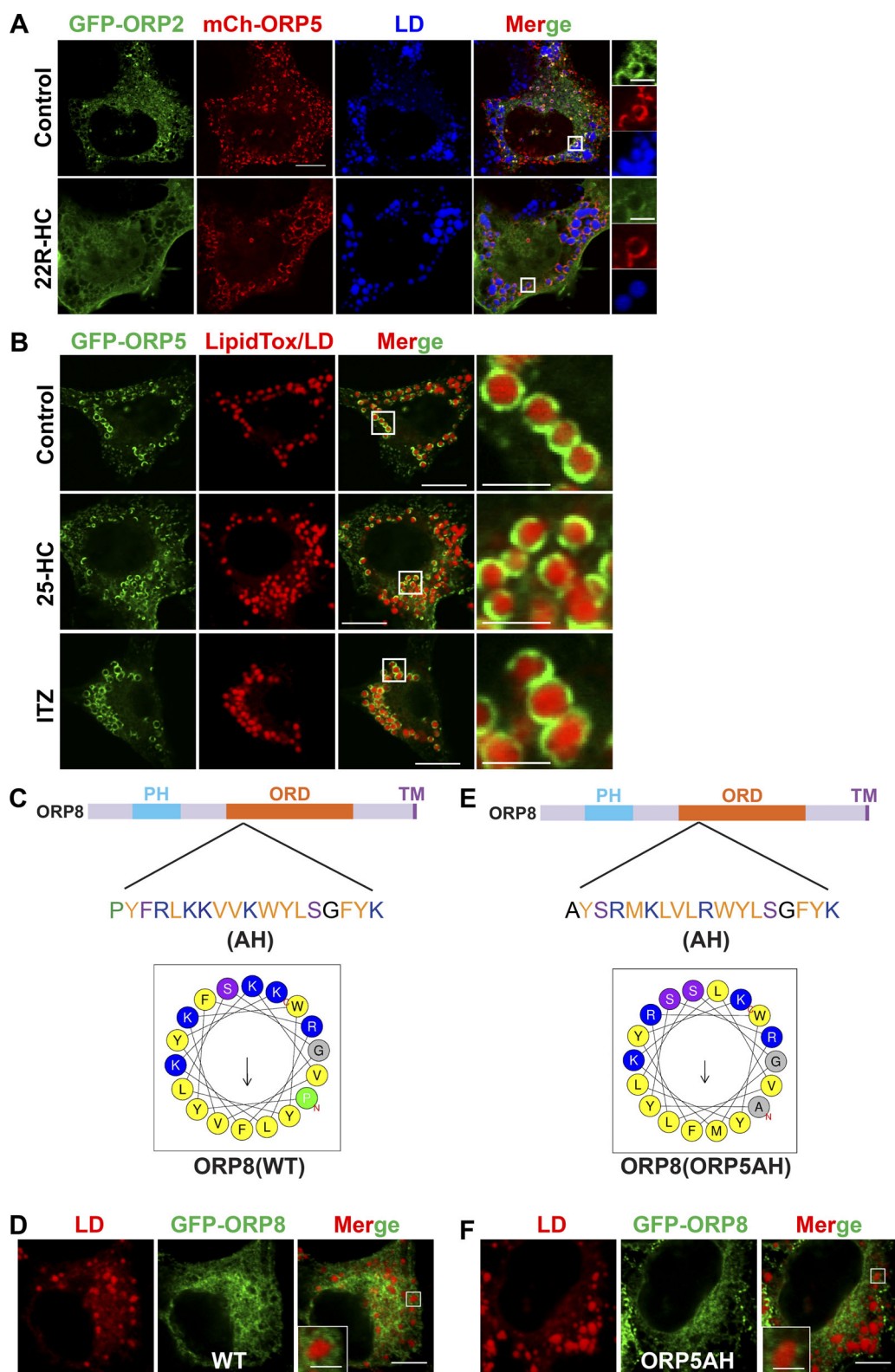

Figure S3.   **Characterization of LD targeting of ORP5 and the role of the AH. (A)** Huh7 cells were transfected with GFP-ORP2 and mCherry-ORP5 for 24 h and treated with oleate in the absence or presence of 22R-hydroxycholesterol (5 µM) for 16 h. Bar = 10 µm (inlay, 2.5 µm). **(B)** HeLa cells were transfected with GFP-ORP5 for 24 h and treated with oleate in the absence or presence of 25-hydroxycholesterol (1 µM) or itraconazole (ITZ, 10 µM) for 16 h. Bars = 10 µm (inlay, 2.5 µm). **(C)** Helical wheel representation of the AH within ORP8ORD generated at HeliQuest. **(D)** HeLa cells were transfected with GFP-ORP8 for 24 h and treated with oleate in the for 16 h. Bar = 10 µm (inset, 2.5 µm). **(E)** Helical wheel representation of ORP8AH substituted with ORP5AH generated at HeliQuest. **(F)** HeLa cells were transfected with GFP-ORP8 containing ORP5AH for 24 h and treated with oleate for 16 h. Bar = 10 µm (inset, 2.5 µm).

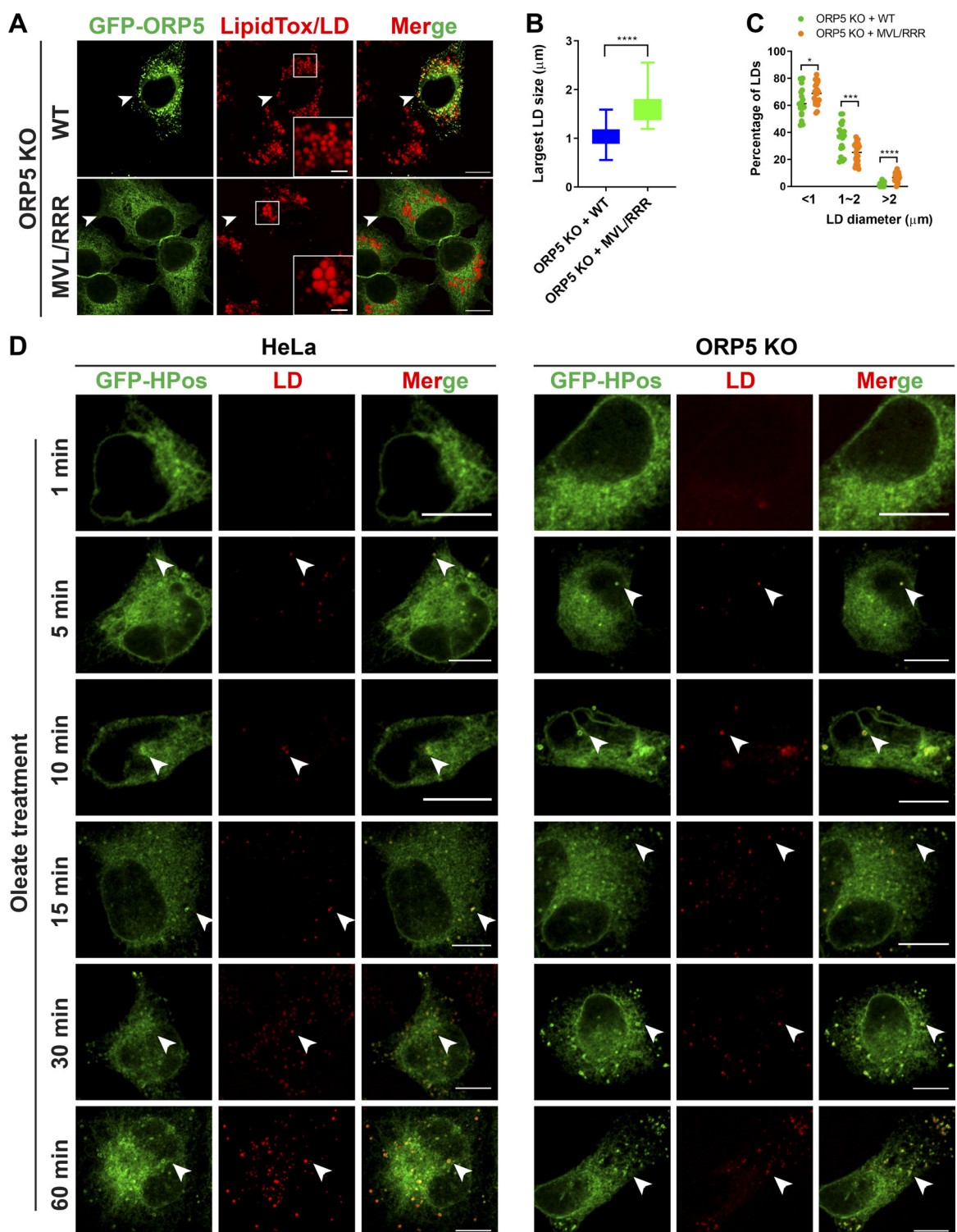

Figure S4. **Role of ORP5AH in LD size regulation and the effect of ORP5 deficiency on early LD formation. (A)** Confocal images of oleate-treated ORP5KO cells expressing GFP-ORP5 or the AH mutant, GFP-ORP5 (MVL/RRR). Arrowheads indicate transfected cells. Bars = 10 µm (insets, 2.5 µm). **(B)** Sizes of the largest LDs in transfected cells in A. Mean ± SD. ****, P < 0.0001; n = 15–20. **(C)** Percentage of LDs with different diameters (<1, 1 to ~2, and >2 µm) in oleate-treated ORP5KO cells expressing GFP-ORP5 or the AH mutant, GFP-ORP5 (MVL/RRR). Mean ± SD. *, P < 0.05; ***, P < 0.001; ****, P < 0.0001; n = 20–23. **(D)** WT HeLa or ORP5 KO cells were transfected with GFP-Hpos for 24 h, starved in serum-free medium overnight, then chased with oleate for 1, 5, 10, 15, 30, and 60 min. LDs were stained with LipidTox DeepRed. Arrowheads indicate LDs. Bars = 10 µm.

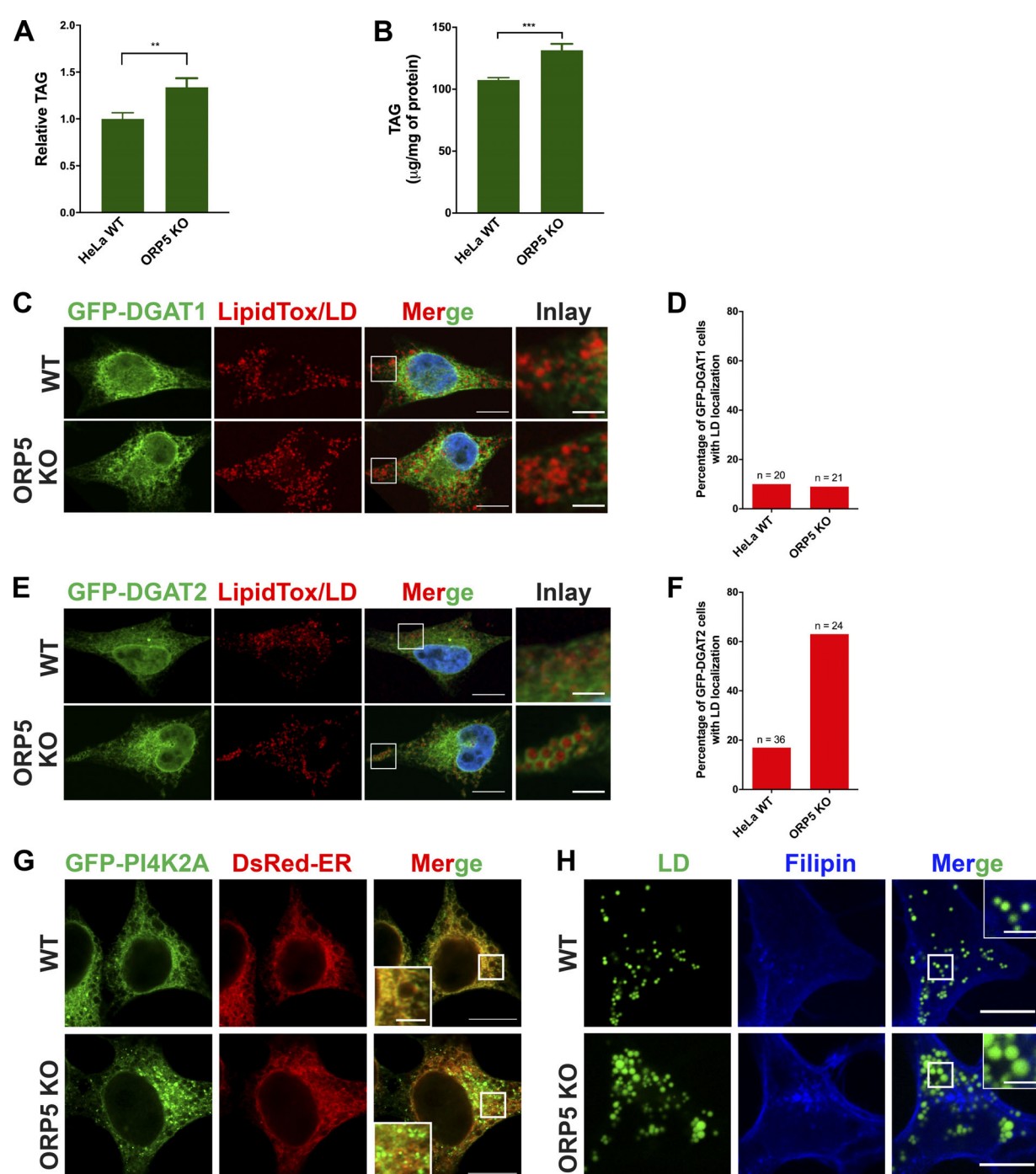

Figure S5. **Effect of ORP5 deficiency on TAG synthesis, localizations of DGAT1/2 and PI4K2A, and cholesterol distribution. (A)** Relative TAG content in HeLa WT and ORP5 KO cells. Cells were treated with oleate for 4 h, and neutral lipids were extracted with hexane and isopropanol, resolved by TLC, and stained with iodine. Densitometry was used to determine the relative TAG content. Mean ± SD. **, P < 0.01, n = 4. Data are representative of three independent experiments with similar results. **(B)** TAG content in HeLa WT and ORP5 KO cells treated as in A was quantified using a fluorometric assay kit. Mean ± SD. ***, P < 0.001, n = 6. Data are representative of two independent experiments with similar results. **(C)** HeLa cells were transfected with GFP-DGAT1 for 24 h and treated with oleate for 4 h. Bars = 10 µm (inlay, 2.5 µm). **(D)** Percentage of transfected HeLa WT and ORP5 KO cells expressing GFP-DGAT1 that show LD localizations. **(E)** HeLa cells were transfected with GFP-DGAT2 for 24 h and treated with oleate for 4 h. Bars = 10 µm (inlay, 2.5 µm). **(F)** Percentage of transfected HeLa WT and ORP5 KO cells expressing GFP-DGAT2 that show LD localizations. **(G)** WT HeLa or ORP5 KO cells were transfected with GFP-PI4K2A and DsRed-ER for 24 h. Bars = 10 µm (insets, 2.5 µm). **(H)** WT HeLa or ORP5 KO cells were treated with oleate for 16 h, followed by filipin staining for free cholesterol and BODIPY staining for LDs. Bars = 10 µm (insets, 2.5 µm).

