## [Reviewer comments · The Journal of Cell Biology]

ORP5 Localizes to ER-Lipid Droplet Contacts and Regulates the Level of PI(4)P on Lipid Droplets

Ximing Du, Linkang Zhou, Yvette Aw, Hoi Yin Mak, Yanqing Xu, James Rae, Wenmin Wang, Armella Zadoorian, Sarah Hancock, Brenna Osborne, Xiang Chen, Jia-Wei Wu, Nigel Turner, Robert Parton, Peng Li, and Hongyuan Yang

Corresponding Author(s): Hongyuan Yang, University of New South Wales and Peng Li, Tsinghua University

Review Timeline:

Submission Date:	2019-05-22
Editorial Decision:	2019-06-17
Revision Received:	2019-09-17
Editorial Decision:	2019-09-30
Revision Received:	2019-10-02

Monitoring Editor: William Prinz

Scientific Editor: Melina Casadio

Transaction Report:

DOI: <https://doi.org/10.1083/jcb.201905162>

June 17, 2019

Re: JCB manuscript #201905162

Prof. Hongyuan Yang
University of New South Wales
School of Biotechnology and Biomolecular Sciences
Sydney 2052
Australia

Dear Prof. Yang,

Thank you for submitting your manuscript entitled "ORP5 Regulates the Level of Phosphatidylinositol-4-Phosphate on Lipid Droplets". The manuscript was assessed by expert reviewers, whose comments are appended to this letter. We sincerely apologize for the delay in communicating this decision to you. We invite you to submit a revision if you can address the reviewers' key concerns, as outlined here.

You will see that the reviewers - #1 and #2 in particular - found the discoveries that ORP5 is present at ER-LD contacts and transports PS/PI4P, documenting PI4P on LDs for the first time, interesting and exciting. These experts in the fields covered by the study however raised experimental and technical comments that we editorially found valid and important to resolve in revision. They provided direct and constructive suggestions to strengthen the work. We strongly encourage you to tackle the reviewers' points, in particular by addressing Rev#2's technical points and by strengthening the localization data, which is an important part of your model (Rev#1 point #1, Rev#3 points #6, 9). Please also address Rev#3's doubts about the lipid distributions and LD phenotypes (#1-2, please explain #4 and address #5). The concerns of Rev#3 about the probes are important but may be challenging to address experimentally. Multiple probes could be used. In addition, changes in PS levels on LDs can be directly assessed by purifying LDs and determining PS levels. At a minimum, the concerns about the probes should be addressed in the discussion. We also agree with Rev#3 that the method used to measure TAG is not really quantitative. A quantitative method should be used. Another important aspect of the revision will be to bolster the conclusions related to the amphipathic helix (Rev#1 #3, Rev#3 #7). Please also tackle Rev#1 point #2 and the more minor technical points from Revs#1 and #3.

Addressing the questions about the broader implications/significance of the findings (Rev#2, points #5-6; Rev#3, point #3) would be interesting to deepen the biological analysis, but data addressing these points will not be necessary for the work to be appropriate for JCB. Discussion of these points would be needed if no data are available.

Please let us know if you have any questions about the revision or anticipate any issues addressing the reviewers' points. We would be happy to discuss the revisions further as needed. While you are revising your manuscript, please also attend to the following editorial points to help expedite the publication of your manuscript. Please direct any editorial questions to the journal office.

GENERAL GUIDELINES:

Text limits: Character count for an Article is < 40,000, not including spaces. Count includes title page, abstract, introduction, results, discussion, acknowledgments, and figure legends. Count does not include materials and methods, references, tables, or supplemental legends.

Figures: Articles may have up to 10 main text figures. Figures must be prepared according to the policies outlined in our Instructions to Authors, under Data Presentation, <http://jcb.rupress.org/site/misc/ifora.xhtml>. All figures in accepted manuscripts will be screened prior to publication.

IMPORTANT: It is JCB policy that if requested, original data images must be made available. Failure to provide original images upon request will result in unavoidable delays in publication. Please ensure that you have access to all original microscopy and blot data images before submitting your revision.

Supplemental information: There are strict limits on the allowable amount of supplemental data. Articles may have up to 5 supplemental figures. Up to 10 supplemental videos or flash animations are allowed. A summary of all supplemental material should appear at the end of the Materials and methods section.

The typical timeframe for revisions is three months; if submitted within this timeframe, novelty will not be reassessed at the final decision. Please note that papers are generally considered through only one revision cycle, so any revised manuscript will likely be either accepted or rejected.

Thank you for this interesting contribution to the Journal of Cell Biology. You can contact us at the journal office with any questions, cellbio@rockefeller.edu or call (212) 327-8588.

Sincerely,

William Prinz, PhD
Monitoring Editor, Journal of Cell Biology

Melina Casadio, PhD
Senior Scientific Editor, Journal of Cell Biology

Reviewer #1 (Comments to the Authors (Required)):

Du et al. report research that links ORP5 (OSBP-related-protein 5) to lipid droplet (LD) function. The ORPs typically transfer lipids between bilayers. The authors asked if any of them are involved in lipid transfer from/to LDs. First, the group performed a screen by overexpressing 11 ORPs in cells grown with oleate. Only ORP5 and ORP5B significantly localized to droplets, as well as to plasma membrane. EM-APEX and organelle fractionation confirmed this localization. They showed that the ORD (OSBP-related domain) is necessary and sufficient for droplet targeting, while the PH

domain targets to the plasma membrane. Furthermore, an amphipathic helix in the ORD was found to be necessary for LD targeting. To assess effects on LD function, ORP5-knockdown and knockout cells were made. The lack of ORP5 resulted in larger droplets, stronger targeting of DGAT2 to droplets, and an increase in cellular triglycerides. The effect was rescued by wild type but not mutations in PI-binding or PS-binding sites, which also failed to localize correctly. Lack of ORP5 also lead to more PI4P, and less PS, on droplets, suggesting a role of ORP5 in translocating PS to LD membranes at the expense of PI4P. Furthermore, these cells had more PI4K2A on droplets.

This report convincingly shows that ORP5 can target to ER/LD junctions and control levels of PI4P and PS on droplets, as well as cellular levels of TG and lipid droplet size. The droplet targeting domain involves the hydrophobic face of a amphipathic helix. This is the first report to my knowledge of an ORP functioning on lipid droplets. It should be of significant interest to readers of the journal.

However, I have some concerns that should be addressed in a revision:

(1) While overexpressed protein clearly can target to droplets, as shown by beautiful APEX-EM data, the targeting of the endogenous is less clear, and I find the phenotype of the knockout/knockdown experiments more convincing than the localization data shown in Fig. 2G, regarding function of ORP5. The pattern of ORP5 and BODIPY puncta seem pretty random and independent in Fig. 2G. Is there more colocalization here than would occur randomly? Repeating the APEX-EM experiment on the endogenous protein would be very helpful to support the claim of co-localization of the endogenous protein.

(2) As droplets are larger and TAG is increased in KO/KD cells compared to wildtype, more of a molecule (such as PIP in Fig 6B) may be associated with droplets based only on the increase in surface area of droplets in KO/KD cells. Is this controlled for?

(3) Fig. 4: Clearly the amphipathic helix (AH) identified in the ORD is necessary for binding of ORP5 to droplets, but it may be acting indirectly (in stabilizing the ORD structure). Have the authors modeled the ORD in ORP5 to the other ORDs in the database in which structures have been determined? Is it predicted to be exposed on the surface making it accessible to a droplet? Can the AH, conjugated to GFP, for example, bind to droplets, i.e., is it sufficient for binding in isolation?

Other issues:

(4) Fig. 1G: most of the fluorescence is not recovered in the FRAP experiment. Does this imply that most of ORP5 is not exchangeable? I think this deserves a comment.

(5) Fig. 3D: The Results text mentions perilipin 2/ADRP, but the figure shows CGI-58.

Reviewer #2 (Comments to the Authors (Required)):

ORP5 Regulates the Level of Phosphatidylinositol-4-Phosphate on Lipid Droplets
Ximing Du^{1, #, *}, Linkang Zhou^{2, #}, Yvette Aw¹, Hoi Yin Mak¹, James Rae³, Wenmin Wang², Armella Zadoorian¹, Sarah E. Hancock⁴, Nigel Turner⁴, Robert G. Parton³, Peng Li^{2, *} and Hongyuan Yang^{1, *}

Summary:

The manuscript by Du et al describes a mini-screen where they found that ORP5 can localize to ER-LD contact sites upon oleate loading. They show that ORP5 interacts with the LDs through an amphipathic helix within its ligand binding domain. Further, that ORP5 is important to deliver PS to

LDs by transferring PI4P that is generated by PI4K2A. These findings would be of considerable interest to those working in the field of phosphoinositide biology.

Specific Comments:

1. In Figures 3 and 4- ER staining needs to be performed in order to confirm localization
2. In all % localization graphs (Figures 1, 3, 4, 5), can you explain/interpret the exact quantification technique that was applied to calculate the ER and PM, especially when there is no markers/staining for the ER or PM in the figures.
3. In all microscopy performed in paper it is written that n=15-20 cells were counted. Please indicate the replicate. Experiments need to be performed in triplicate, n=3 and statistics then applied.
4. In Figure 7, please provide western blots of PI4K2A knockdown.
5. The authors have shown that oleate loading induces LD formation throughout the paper. Are there physiological stresses that could cause ORP5 lipid transfer at LDs? Or can authors discuss the relevance of their findings to cellular metabolism and/or metabolic disorders or cancers.
6. In light of their recent paper regarding ORP5 and mTORC1 signaling, is there a connection of ORP5 lipid transfer function at LDs in cancer cell proliferation? Is there a role for ORP5 in LD-lysosome interactions?

Typos

1. Page 6 - OPR5 should be ORP5
2. Page 11 - punta should be puncta

In sum, the authors' current conclusion that ORP5 localizes to ER-LD contacts to deliver PS at the expense of PI4P is novel, and most importantly that for the first they show that PI4P is present on LDs.

Reviewer #3 (Comments to the Authors (Required)):

The authors have investigated the interaction and regulation by ORP5 of lipid droplet biogenesis in cultured cells incubated with exogenous oleate. ORP5 is a member of the OSBP/ORP family of lipid transfer proteins that mediate the movement of sterols, PIPs and PS at organelle contact sites. ORP5 was shown to mediate the transfer of PS and PIPs (PI4P and PIP2) at PM-ER contacts, cholesterol transfer at ER-LEL and to localize at mitochondrial contacts where its function is less clear. Evidence in this manuscript shows that overexpressed GFP-tagged ORP5 and a PH-truncated 5B isoform also localizes to the surface of LDs in several cultured cell lines (HeLa cells are the primary model). Light and electron microscopy show that GFP-ORP5 coats the surface of LDs and is present at contacts between LD and the ER. The association is dependent on the ORD and C-terminal TM domain (does not involve the PH domain). Endogenous ORP5 also appears as puncta that are close to LDs but not always in the immediate proximity. Functionally, ORP5 knockout HeLa cells have larger LDs, more associated PI4P and less PS (as determined by biosensor and lipid-specific antibodies). They also show that the PI4K2A isoform is producing PI4P on the LDs and that silencing the kinase causes enlarged LDs. They conclude that ORP5 is responsible for transport of PS to LDs in exchange for PI4P, and this transport pathway is linked to TG storage. The manuscript is clearly written and the conclusions are generally consistent with the reported results.

Major points

1. LDs are known to have very little or no PS and PI4P (based on mass measurements). The study shows that PI4P produced by PI4K2A can be detected with biosensors (P4M) in ORP5 KO cells but not WT cells. It's a novel finding that LDs may contain PI4P but all the PS and PI4P measurements on LDs are made with biosensors that are prone to localized environmental effects. Because of the resolution of images, probes may be reporting PI4P and PS in the ER that surrounds the LD. For example, in Fig. 6A, the PI4P appears in puncta adjacent to LDs, and in Fig. 7E and F the images appear to be in a peri-LD region (that term used in the figure legends) that have irregular patterns. This could represent either the LD surface or closely associated ER.
2. There is unusual PI4P staining in Figure 6A, which shows strong PI4P detection in the Golgi in WT cells but this Golgi staining is absent in the KOs. Given that ORP5 is known to be involved in PIP removal from the PM to the ER, could the PI4P distribution on LDs be a secondary effect of more global disruption of PIP and PS homeostasis due to loss of ORP5 at other contact sites (ie. PM-ER). The authors should have looked more closely at the consequences of ORP5 KO on PI4P and PI45P2 distribution in the entire cell. Both the antibody and P4M probes will detect PI4P in other compartments.
3. What could be the potential function of PI4P and PS on LDs? These are in very low concentrations under steady state conditions, so low that the authors have been unable to detect any mass.
4. The LD phenotype in ORP5K cells is very subtle, with a small increase in LD size and TG mass. It seems that they are selectively focused on the 'largest' LD fraction that is increased in KO cells (Figure 5 and S4). They do not indicate the actual size of this LD fraction or what proportion it makes up of the entire population (figure 7). It could be the >2um fraction, but this is not indicated in the figure or legend.
5. The method used to quantify TAG is unusual. Total lipids are separated by TLC and the TG band is visualized with iodine vapour and quantified by densitometry. Iodine is volatile and immediately dissociates from lipids when removed from a saturated environment. Moreover, iodine staining is sensitive to double bond content and is used to visualize lipids not quantify them. Should use a quantitative method for TG mass-there are lots available.
6. The description of overexpressed ORP5 association with LD-ER contacts in Figure 2 is confusing. The EMs show that GFP-ORP5 extensively coats the surface of most LDs, effectively wrapping the LD in a layer of ER. This suggests that under overexpression conditions, there are extensive contact sites between ER-anchored ORP5 and the surface of the LD. However, the authors also suggest that ORP5 is at ER-LD junctions. These are difficult to see in the EM images and are minor compared to the massive encirclement of LDs with ORP5-enriched ER.
7. The identification of the AH as a potential LD interacting domain is interesting but does not provide conclusive evidence of ORP5 LD interaction domains. Since the ORD is required for LD interaction (fig. 3), mutations in the AH could negatively affect ORD structure and thus prevent LD interaction (ie. a global effect on ORD function like lipid binding). Also unclear whether the AH is a lipid or protein interaction motif.
8. The immunoblot in Fig. 3D should be repeated with reduced protein load in the pellet fraction; it is overloaded and difficult to assess the relative amount of GFP-ORP5 in the LD fraction.
9. The absence of any ER markers in the LD fraction in Fig. 3 also suggests that GFP-ORP5 is not

tethered strongly to LDs or that the ER is stripped off during isolation (see #2 above). This is surprising since the images of GFP-ORP5 association with LDs in Fig.1 and 2 show it covering the surface of the LD.

10. Show quantitation of LD distribution in Fig.5 C and F. Are the changes significant?

Editors' comments:

You will see that the reviewers - #1 and #2 in particular - found the discoveries that ORP5 is present at ER-LD contacts and transports PS/PI4P, documenting PI4P on LDs for the first time, interesting and exciting. These experts in the fields covered by the study however raised experimental and technical comments that we editorially found valid and important to resolve in revision. They provided direct and constructive suggestions to strengthen the work. We strongly encourage you to tackle the reviewers' points, in particular by addressing Rev#2's technical points and by strengthening the localization data, which is an important part of your model (Rev#1 point #1, Rev#3 points #6, 9). Please also address Rev#3's doubts about the lipid distributions and LD phenotypes (#1-2, please explain #4 and address #5). The concerns of Rev#3 about the probes are important but may be challenging to address experimentally. Multiple probes could be used. In addition, changes in PS levels on LDs can be directly assessed by purifying LDs and determining PS levels. At a minimum, the concerns about the probes should be addressed in the discussion. We also agree with Rev#3 that the method used to measure TAG is not really quantitative. A quantitative method should be used. Another important aspect of the revision will be to bolster the conclusions related to the amphipathic helix (Rev#1 #3, Rev#3 #7). Please also tackle Rev#1 point #2 and the more minor technical points from Revs#1 and #3.

Addressing the questions about the broader implications/significance of the findings (Rev#2, points #5-6; Rev#3, point #3) would be interesting to deepen the biological analysis, but data addressing these points will not be necessary for the work to be appropriate for JCB. Discussion of these points would be needed if no data are available.

Many thanks for your comments. We have addressed each of the reviewers' points as we outline below.

Reviewer #1

Du et al. report research that links ORP5 (OSBP-related-protein 5) to lipid droplet (LD) function. The ORPs typically transfer lipids between bilayers. The authors asked if any of them are involved in lipid transfer from/to LDs. First, the group performed a screen by overexpressing 11 ORPs in cells grown with oleate. Only ORP5 and ORP5B significantly localized to droplets, as well as to plasma membrane e. EM-APEX and organelle fractionation confirmed this localization. They showed that the ORD (OSBP-related domain) is necessary and sufficient for droplet targeting, while the PH domain targets to the plasma membrane. Furthermore, an amphipathic helix in the ORD was found to be necessary for LD targeting. To assess effects on LD function, ORP5-knockdown and knockout cells were made. The lack of ORP5 resulted in larger droplets, stronger targeting of DGAT2 to droplets, and an increase in cellular triglycerides. The effect was rescued by wild type but not mutations in PI-binding or PS-binding sites, which also failed to localize correctly. Lack of ORP5 also lead to more PI4P, and less PS, on droplets, suggesting a role of ORP5 in translocating PS to LD membranes at the expense of PI4P. Furthermore, these cells had more PI4K2A on droplets.

This report convincingly shows that ORP5 can target to ER/LD junctions and control levels

of PI4P and PS on droplets, as well as cellular levels of TG and lipid droplet size. The droplet targeting domain involves the hydrophobic face of an amphipathic helix. This is the first report to my knowledge of an ORP functioning on lipid droplets. It should be of significant interest to readers of the journal.

Many thanks for recognizing the significance of this work.

However, I have some concerns that should be addressed in a revision:

(1) While overexpressed protein clearly can target to droplets, as shown by beautiful APEX-EM data, the targeting of the endogenous is less clear, and I find the phenotype of the knockout/knockdown experiments more convincing than the localization data shown in Fig. 2G, regarding function of ORP5. The pattern of ORP5 and BODIPY puncta seem pretty random and independent in Fig. 2G. Is there more colocalization here than would occur randomly? Repeating the APEX-EM experiment on the endogenous protein would be very helpful to support the claim of co-localization of the endogenous protein.

Figure R1. (A) HeLa cells were treated with oleate overnight and then processed for immunofluorescence of endogenous ORP5 and BODIPY staining of LDs. One representative merged image with ORP5 and BODIPY channels is shown in the left. The pattern of ORP5 and BODIPY puncta within the inlay is compared upon the 90° rotation of the ORP5 channel. Bar = 10 μ m (inlay, 2 μ m). (B) Comparison of the percentages of LDs associated with ORP5 between images with normal and rotated ORP5 channels.

We repeated the ORP5 immunofluorescence experiment as shown in Fig. 2G to carefully examine whether the association between endogenous ORP5 and BODIPY-stained LDs is random. We calculated the percentage of LDs associated with ORP5 puncta from merged images composed of the two channels of ORP5 and LDs. We then calculated the percentage

again from the same image with the channel of ORP5 rotated for 90° (Figure R1A). If the association between LDs and endogenous ORP5 is random, the two percentage values should be similar. Based on all images analysed, ~20% of LDs were found to associate with ORP5 puncta. However, this number dropped to ~8% when the ORP5 channel was rotated for 90° in the same images (Figure R1B). This analysis strongly suggests that there is a specific, but not random, connection between endogenous ORP5 and LDs. Due to the space limit, we include these results here in Figure R1 for your reference.

The APEX-EM experiment requires a GFP to be fused to the protein of interest (see Figure 2A). To repeat the APEX-EM experiment on the endogenous ORP5, we tried the CRISPR/Cas9 system to tag the endogenous ORP5 with GFP. We have successfully generated the ORP5 guide RNA/Cas9 construct and a donor plasmid containing GFP flanked by recombinant arms homologous to targeted *ORP5* genomic sequence. We have carried out three rounds of clonal screen (genomic DNA PCR and Western blot analysis) to select positive cell clones with endogenous GFP-tagged ORP5. However, we have yet to succeed in this screen. At least in HeLa cells, CRISPR tagging of the endogenous ORP5 locus with GFP appears to be very challenging. We sincerely seek your understanding on this.

(2) As droplets are larger and TAG is increased in KO/KD cells compared to wildtype, more of a molecule (such as PIP in Fig 6B) may be associated with droplets based only on the increase in surface area of droplets in KO/KD cells. Is this controlled for?

Thanks for the suggestion. We previously showed that knocking-down CDP-Diacylglycerol synthase 2 (CDS2) caused the formation of giant/“supersized” LDs (Qi et al, JLR, 2016;

Figure R2 Localization of the PI4P probe GFP-P4M in CDS2 knockdown cells. (A) Western Blot analysis of CDS2 in HeLa cells treated with control or CDS2 siRNAs. (B) Confocal images of siRNA transfected HeLa cells expressing the PI(4)P sensor GFP-P4M treated with oleate. Bars = 10 μ m. (C) Quantitation of GFP-P4M surrounding LDs in (B). AU, arbitrary unit. ****, $p < 0.0001$, $n = 16$.

PMID: 26946540). We used CDS2 knockdown cells as a control and examined the localization of the PI4P probe, GFP-P4M, in these cells. The increase of LD sizes in CDS2 knockdown cells did not promote the association of GFP-P4M with the LDs. Due to the space limit, we included the results here for your reference (Figure R2).

(3) Fig. 4: Clearly the amphipathic helix (AH) identified in the ORD is necessary for binding of ORP5 to droplets, but it may be acting indirectly (in stabilizing the ORD structure). Have the authors modeled the ORD in ORP5 to the other ORDs in the database in which structures have been determined? Is it predicted to be exposed on the surface making it accessible to a droplet? Can the AH, conjugated to GFP, for example, bind to droplets, i.e., is it sufficient for binding in isolation?

This is a great point. Thanks. We showed in the original manuscript that the putative AH domain of ORP5 was necessary but not sufficient for LD targeting. As suggested, we modelled ORP5-ORD with that of yeast Osh6p, which is a close homologue of mammalian ORP5, and found the putative AH domain in a region that appears to be important for maintaining the ORD structure (Figure R3, red). Based on the modelling, this AH domain appears to be important for maintaining the overall structure of the ORD, although it does not and cannot completely rule out the possibility that the AH domain might rotate upon interacting with LDs.

As suggested, we have also fused the ORP5AH (a.a. 422-439) domain with GFP and tested its localization in cells treated with oleate. This fusion protein failed to show a clear LD

Figure R3. Modelling of the ORP5-ORD to the yeast Osh6p. The AH domain is in red.

pattern. Interestingly however, when the same GFP-ORP5AH was anchored to the ER by the transmembrane region of ORP5, it can now localize to LDs (see new Figure 4F). Please note that without the ER anchor, LD targeting by full length ORD5 was also dramatically reduced (Figure 3C). These results suggest that the AH domain alone in ORP5-ORD can recognize and interact with LDs.

Based on these results and a lack of ORP5-ORD structure, we have now added the following statement in the discussion: “However, it is important to note that the AH domain may also

be important for stabilizing the ORP5-ORD based on modelling with the ORD of Osh6p (data not shown). Future structural and functional analyses of ORP5-ORD are needed to determine the precise mechanism by which ORP5 is targeted to the LDs” (page 15).

Other issues:

(4) *Fig. 1G: most of the fluorescence is not recovered in the FRAP experiment. Does this imply that most of ORP5 is not exchangeable? I think this deserves a comment.*

The incomplete recovery of the GFP signals may be due to the presence of an immobile fraction of GFP-ORP5 anchored to the ER membranes. As shown in Fig. 1F, most of the fluorescence on the LD surface was recovered within two minutes. As suggested, we have added the sentence “The incomplete recovery of the GFP signals five minutes after photobleaching may be due to the presence of an immobile fraction of GFP-ORP5 anchored to the ER membranes” to the main text (page 7).

(5) *Fig. 3D: The Results text mentions perilipin 2/ADRP, but the figure shows CGI-58.*

GFP-ADRP was transfected in cells as a positive control for LD fractionation studies. CGI-58 was used as a marker to label the LD fractions in the immunoblotting experiments.

Reviewer #2:

ORP5 Regulates the Level of Phosphatidylinositol-4-Phosphate on Lipid Droplets
*Ximing Du1, #, *, Linkang Zhou2, #, Yvette Aw1, Hoi Yin Mak1, James Rae3, Wenmin Wang2, Armella Zadoorian1, Sarah E. Hancock4, Nigel Turner4, Robert G. Parton3, Peng Li2, * and Hongyuan Yang1, **

Summary:

The manuscript by Du et al describes a mini-screen where they found that ORP5 can localize to ER-LD contact sites upon oleate loading. They show that ORP5 interacts with the LDs through an amphipathic helix within its ligand binding domain. Further, that ORP5 is important to deliver PS to LDs by transferring PI4P that is generated by PI4K2A. These findings would be of considerable interest to those working in the field of phosphoinositide biology.

Many thanks for recognizing the importance of this work.

Specific Comments:

1. In Figures 3 and 4- ER staining needs to be performed in order to confirm localization

We co-transfected Hela cells with the ER marker, ER-DsRed, and repeated the experiments. Figures 3 and 4 have been revised accordingly.

2. In all % localization graphs (Figures 1, 3, 4, 5), can you explain/interpret the exact quantification technique that was applied to calculate the ER and PM, especially when there is no markers/staining for the ER or PM in the figures.

The localization of ER-anchored GFP-ORP5/8 to the ER and/or ER-PM junctions has been well documented in our previous studies (Du et al, JCB, 2011; Ghai et al, Nat. Comms., 2019). ORP5 predominantly accumulates at the ER-PM junctions while ORP8 mainly localizes to tubular ER. Please note that we used the MAPPER construct as an ER-PM junction marker (Change et al, Cell Reports, 2013) in Figure 1C and GFP-Sec61 β as an ER maker in Figure 1D to confirm these previous findings. For quantification, we examined those transfected cells and characterized the localization pattern of ORP5/8 (ER-PM, ER, and LD) based on their association with MAPPER, GFP-Sec61 β , and LDs stained with the florescent dyes. The % localization in these figures represented the percentage of transfected cells showing different localization pattern. Since the localization of ORP5 to LDs and ER/ER-PM junctions was established and confirmed in Figure 1, the subsequent quantification performed in Figures 3, 4, and 5 was carried out in the same way.

In order to minimize the ambiguity, we have now only focused on the phenotype of LD targeting and simplified the quantification by calculating the percentage of transfected cells showing the LD targeting phenotype. Subsequently, we have modified all % localization graphs in Figures 1E, 3C, 4E, and 5O, as well as those in the supplemental figures (Figures S1B, S5D, S5F).

3. In all microscopy performed in paper it is written that $n=15-20$ cells were counted. Please indicate the replicate. Experiments need to be performed in triplicate, $n=3$ and statistics then applied.

All of data were representative from at least three independent experiments with similar results. We added the sentence stating that all results were representative of at least 3 experiments in the figure legends.

4. In Figure 7, please provide western blots of PI4K2A knockdown.

We performed the immunoblotting experiment using a polyclonal anti-PI4K2A antibody and confirmed the knockdown efficiency. The results have been included in the revised Figure 7 (7E).

5. The authors have shown that oleate loading induces LD formation throughout the paper. Are their physiological stresses that can would cause ORP5 lipid transfer at LDs? Or can authors discuss the relevance of their findings to cellular metabolism and/or metabolic disorders or cancers.

A key question in LD biology is how LDs expand upon lipid loading, which is a common metabolic stress given the prevalence of obesity and related metabolic disorders. As LDs are enclosed by a monolayer of phospholipids, they need to acquire phospholipids to grow. Moreover, they need to acquire a variety of phospholipid species since each type of phospholipid may have a distinct function. As stated in the discussion, VPS13 proteins may

mediate the bulk transfer of lipids to the growing LDs. ORP5, as we show here, mediates the selective transfer of PS to LDs at the expense of PI4P. How the ORP5 pathway matters in physiology/metabolism is currently unknown. There are no reported ORP5^{-/-} mouse models and we are in the process of making one. As for cancer, the pool of ORP5 at ER-LD junction may play a minor role since cancer signalling and mTOR activation may primarily occur at or near the plasma membrane (see below).

6. In light of their recent paper regarding ORP5 and mTORC1 signaling, is there a connection of ORP5 lipid transfer function at LDs in cancer cell proliferation? Is there role for ORP5 in LD-lysosome interactions?

This is a good question. ORP5 has been observed at multiple cellular contact sites. This is not surprising since most organelles need PS. However, it is not clear which PS transfer activity is essential to cancer cell proliferation. We favour the activity at ER-PM contact sites because the PS at plasma membrane is known to be required for AKT signalling (see Huang, B.X., M. Akbar, K. Kevala, and H.Y. Kim. 2011. Phosphatidylserine is a critical modulator for Akt activation. *The Journal of cell biology*. 192:979-992). Moreover, the interaction between ORP5 and mTOR could occur on peripheral lysosomes, which are close to cell surface under nutrient-rich conditions. LDs can indeed interact with lysosomes, but such interactions often occur under starved conditions that favour autophagy/lipophagy. LD-lysosome interaction was not obvious under the nutrient-rich conditions we used.

This paper focuses on ORP5 and PIPs on LD surface. Whether ORP5's role in LD function is related to cancer progression, and whether ORP5 regulates LD-lysosome interactions will be important directions for future research.

Typos

- 1. Page 6 - OPR5 should be ORP5*
- 2. Page 11 - punta should be puncta*

Thanks. Corrected.

In sum, the authors' current conclusion that ORP5 localizes to ER-LD contacts to deliver PS at the expense of PI4P is novel, and most importantly that for the first they show that PI4P is present on LDs.

Thanks again.

Reviewer #3:

The authors have investigated the interaction and regulation by ORP5 of lipid droplet biogenesis in cultured cells incubated with exogenous oleate. ORP5 is a member of the OSBP/ORP family of lipid transfer proteins that mediate the movement of sterols, PIPs and PS at organelle contact sites. ORP5 was show to mediate the transfer of PS and PIPs (PI4P and PIP2) at PM-ER contacts, cholesterol transfer at ER-LEL and to situate at mitochondrial contacts where its function is less clear. Evidence in this manuscript shows that

overexpressed GFP-tagged ORP5 and a PH-truncated 5B isoform also localizes to the surface of LDs in several cultured cell lines (HeLa cells are the primary model). Light and electron microscopy show that GFP-ORP5 coats the surface of LDs and is present at contacts between LD and the ER. The association is dependent on the ORD and C-terminal TM domain (does not involve the PH domain). Endogenous ORP5 also appears as puncta that are close to LDs but not always in the immediate proximity. Functionally, ORP5 knockout HeLa cells have larger LDs, more associated PI4P and less PS (as determined by biosensor and lipid-specific antibodies). They also show that the PI4K2A isoform is producing PI4P on the LDs and that silencing the kinase causes enlarged LDs. They conclude that ORP5 is responsible for transport in PS to LDs in exchange for PI4P, and this transport pathway is linked to TG storage. The manuscript is clearly written and the conclusions are generally consistent with the reported results.

Major points

1. LDs are known to have very little or no PS and PI4P (based on mass measurements). The study shows that PI4P produced by PI4K2A can be detected with biosensors (P4M) in ORP5 KO cells but not WT cells. It's a novel finding that LDs may contain PI4P but all the PS and PI4P measurements on LDs are made with biosensors that are prone to localized environmental effects. Because of the resolution of images, probes may be reporting PI4P and PS in the ER that surrounds the LD. For example, in Fig. 6A, the PI4P appears in puncta adjacent to LDs, and in Fig. 7E and F the images appear to be in a peri-LD region (that term used in the figure legends) that have irregular patterns. This could represent either the LD surface or closely associated ER.

We admit that the sensors have limitations in detecting the localization of cellular lipids. However, there are currently no better alternatives. As for the possibility of PI4P or PS on the ER but not LDs, please take the following into account: a) We only detected PI4P-LD association in the absence of ORP5. Importantly, ORP5 is known to bring PI4P back to the ER, so the depletion of ORP5 should decrease, but not increase PI4P on the ER. Therefore, it is highly unlikely that the PI4P detected in the absence of ORP5 is on the ER. b) As for PS, it was very well established that PS localize to the luminal side of the ER (Fairn et al, *JCB*, 2011; PMID: 21788369) and there is little, if any, PS on the cytoplasmic leaflet of the ER under steady state. Thus, the PS detected in this study is unlikely to be on the ER since the sensor we used exists only in the cytosol.

2. There is unusual PI4P staining in Figure 6A, which shows strong PI4P detection in the Golgi in WT cells but this Golgi staining is absent in the KOs. Given that ORP5 is known to be involved in PIP removal from the PM to the ER, could the PI4P distribution on LDs be a secondary effect of more global disruption of PIP and PS homeostasis due to loss of ORP5 at other contact sites (ie. PM-ER). The authors should have looked more closely at the consequences of ORP5 KO on PI4P and PI45P2 distribution in the entire cell. Both the antibody and P4M probes will detect PI4P in other compartments.

We have previously shown that there are no major changes in PM PI4P in ORP5 KD cells (Ghai et al, *Nat Comm*, 2017. PMID 28970484). We have now further examined the Golgi localization of PI4P and revised Figure 6A accordingly. In the majority of ORP5 deficient

cells, the distribution of PI4P to the Golgi was not affected. Even in some WT cells, the PI4P staining of the Golgi was not distinct. Thus, the variability of Golgi PI4P pattern may be attributed to some other factors such as cell cycle, but not a loss of ORP5. Importantly, there

Figure R4. PI4P positive Golgi regions in HeLa wild-type and ORP5 KO cells. (A) Immunofluorescence of PI(4)P in oleate treated HeLa wild-type and ORP5 KO cells. LDs were stained with BODIPY. Bars = 10 μ m. (B) Percentage of cells with PI4P stained Golgi regions in HeLa wild-type and ORP5 KO cells.

are LD-associated PI4P in ORP5 KO cells where distinct Golgi PI4P pattern is apparent (Figure R4). Therefore, it is highly unlikely that the LD-associated PI4P is secondary to global PI4P changes of the cell. Due to the space limit, we included these results here for your reference only.

3. What could be the potential function of PI4P and PS on LDs? These are in very low concentrations under steady state conditions, so low that the authors have been unable to detect any mass.

This is a good question. From our observation that PI4P is readily observed on LD surface only in ORP5-deficient cells, PI4P is likely to be primarily needed for the transport function of ORP5, and maybe also that of ORP2. OSBP and ORPs can consume a large quantity of PIPs (see PMID: 30581148; 28978670), and the consumption depends on the amount of bulk lipids such as cholesterol to be transported. In the absence of ORP5, the consumption of LD-associated PI4P is reduced, allowing its detection.

As for PS, given its minute amount on LD surface, it is unlikely to play a structural role or impact the overall charge of the LDs. We tried a fluorescence membrane charge sensor (Ma et al, Nat Biotechnol. 2017; PMID: 28288102) to detect the change of surface charge of LDs in wild-type and ORP5KO cells, but no apparent differences were observed. Most likely the PS on LDs plays a signalling role, but at this stage, we do not know the proteins that may interact with PS on LDs. Importantly, the key points of the current study are the localization of ORP5 to ER-LD contacts and the presence of PI4P on LDs in the absence of ORP5. We

will aim to dissect the role of LD-localized PS in future studies. We sincerely seek your understanding on this.

4. The LD phenotype in ORP5KO cells is very subtle, with a small increase in LD size and TG mass. It seems that they are selectively focused on the 'largest' LD fraction that is increased in KO cells (Figure 5 and S4). They do not indicate the actual size of this LD fraction or what proportion it makes up of the entire population (figure 7). It could be the >2µm fraction, but this is not indicated in the figure or legend.

We have now provided the quantitative data in Figures 5D, 5H, 5K, 5N, 7I, and S4C. These represent statistical analysis of the percentages of three populations of LDs ($\emptyset < 1 \mu\text{m}$, $\emptyset = 1\sim 2 \mu\text{m}$, $\emptyset > 2 \mu\text{m}$) per cell between different groups.

5. The method used to quantify TAG is unusual. Total lipids are separated by TLC and the TG band is visualized with iodine vapour and quantified by densitometry. Iodine is volatile and immediately dissociates from lipids when removed from a saturated environment. Moreover, iodine staining is sensitive to double bond content and is used to visualize lipids not quantify them. Should use a quantitative method for TG mass-there are lots available.

We agree that the method was semi-quantitative and could only indicate a relative change of TAG in WT and ORP5 KO cells. As suggested, we have now used a TAG quantification kit and measured the TAG mass in these cells and normalised the values to total cellular proteins. The results are consistent with the TLC experiment and have been included in the revised Figure S5B.

6. The description of overexpressed ORP5 association with LD-ER contacts in Figure 2 is confusing. The EMs show that GFP-ORP5 extensively coats the surface of most LDs, effectively wrapping the LD in a layer of ER. This suggests that under overexpression conditions, there are extensive contact sites between ER-anchored ORP5 and the surface of the LD. However, the authors also suggest that ORP5 is at ER-LD junctions. These are difficult to see in the EM images and are minor compared to the massive encirclement of LDs with ORP5-enriched ER.

Our understanding is that ER-LD junctions are the same as ER-LD contact sites/contacts. To be consistent, we have now used contact sites/contacts throughout the manuscript.

7. The identification of the AH as a potential LD interacting domain is interesting but does not provide conclusive evidence of ORP5 LD interaction domains. Since the ORD is required for LD interaction (fig. 3), mutations in the AH could negatively affect ORD structure and thus prevent LD interaction (ie. a global effect on ORD function like lipid binding). Also unclear whether the AH is a lipid or protein interaction motif.

Thanks. This is a great point. See our analyses and answers to point 3 by reviewer 1.

8. *The immunoblot in Fig. 3D should be repeated with reduced protein load in the pellet fraction; it is overloaded and difficult to assess the relative amount of GFP-ORP5 in the LD fraction.*

We have now repeated the LD fractionation experiment and revised Figure 3D and 3E accordingly.

9. *The absence of any ER markers in the LD fraction in Fig. 3 also suggests that GFP-ORP5 is not tethered strongly to LDs or that the ER is stripped off during isolation (see #2 above). This is surprising since the images of GFP-ORP5 association with LDs in Fig.1 and 2 show it covering the surface of the LD.*

In general, ER markers are rarely detected in LD fractions, and they cannot and should not be used to label the LD fraction. While ORP5 appears to cover a large portion of LD, the interaction between ORP5 and LDs may not be strong enough to bring the ER membranes to the LD fraction. Since we did not examine every possible ER marker other than calnexin, we cannot rule out that some ER proteins may exist in the LD fraction. Importantly however, the point of Figure 3D is merely to show that ORP5 can associate with LDs by a technique besides microscopy, and the results are clear that ORP5 can be found in the LD fraction. Together with the imaging data, these results support ORP5's association with LDs.

10. *Show quantitation of LD distribution in Fig.5 C and F. Are the changes significant?*

We have now quantified the different sizes of LDs and provided statistical analysis (new Figures 5D and 5H).

September 30, 2019

RE: JCB Manuscript #201905162R

Prof. Hongyuan Yang
University of New South Wales
School of Biotechnology and Biomolecular Sciences
University of New South Wales
Sydney 2052
Australia

Dear Prof. Yang,

Thank you for submitting your revised manuscript entitled "ORP5 Localizes to ER-Lipid Droplet Contacts and Regulates the Level of PI(4)P on Lipid Droplets". You'll see that all the reviewers appreciated the care with which you tackled the revisions. Please consider integrating Rev#3's point in the manuscript text as you prepare your final files. We would be happy to publish your paper in JCB pending final revisions necessary to meet our formatting guidelines (see details below).

1) Text limits: Character count for Articles and Tools is < 40,000, not including spaces. Count includes title page, abstract, introduction, results, discussion, acknowledgments, and figure legends. Count does not include materials and methods, references, tables, or supplemental legends.

2) Titles, eTOC: Please consider the following revision suggestions aimed at increasing the accessibility of the work for a broad audience and non-experts.

Running title (50 characters max, including spaces): the running title should be more descriptive and more precisely encompass the advance.

Suggestion: ORP5 modulates PI(4)P levels on lipid droplets

****Please include both the running title and eTOC summary on the title page of the revised manuscript****

eTOC summary: A 40-word summary that describes the context and significance of the findings for a general readership should be included on the title page. The statement should be written in the present tense and refer to the work in the third person.

****Edits are needed to meet our preferred style****

Revised version:

Lipid droplets (LDs) are important organelles for cell metabolism. Here, Du, Zhou, et al. show that phosphatidylinositol-4-phosphate (PI(4)P) produced by PI4K2A can exist on LDs and is used/consumed by ORP5, which localizes to ER-LD contacts during the growth of LDs.

3) Figure formatting: Scale bars must be present on all microscopy images, including inset magnifications. Please add scale bars to 1BCD (magnifications), 2BD (magnifications), 2G (inlay), 3B

(magnifications), 4CDG (magnifications), 5BFIL (magnifications), 6ACE (magnifications), 7GH (magnifications), S1A (magnifications), S3ABDF (magnifications), S4A (mags), S5CEGH (mags).

4) Statistical analysis: Error bars on graphic representations of numerical data must be clearly described in the figure legend. The number of independent data points (n) represented in a graph must be indicated in the legend. Statistical methods should be explained in full in the materials and methods. For figures presenting pooled data the statistical measure should be defined in the figure legends.

Please indicate n/sample size/how many experiments the data are representative of: 1G, 5M, 7D, S4C

5) Materials and methods: Should be comprehensive and not simply reference a previous publication for details on how an experiment was performed. Please provide full descriptions in the text for readers who may not have access to referenced manuscripts.

- Please include sequences for all siRNAs used, including negative controls if those were made available to you from the manufacturer.
- Please include the basic genetic features for **all** constructs, even if described in other papers or gifted to you by other researchers.
- Microscope image acquisition: The following information must be provided about the acquisition and processing of images:
 - a. Make and model of microscope
 - b. Type, magnification, and numerical aperture of the objective lenses
 - c. Temperature
 - d. imaging medium
 - e. Fluorochromes
 - f. Camera make and model
 - g. Acquisition software
 - h. Any software used for image processing subsequent to data acquisition. Please include details and types of operations involved (e.g., type of deconvolution, 3D reconstitutions, surface or volume rendering, gamma adjustments, etc.).

A. MANUSCRIPT ORGANIZATION AND FORMATTING:

Full guidelines are available on our Instructions for Authors page, <http://jcb.rupress.org/submission-guidelines#revised>. ****Submission of a paper that does not conform to JCB guidelines will delay the acceptance of your manuscript.****

B. FINAL FILES:

-- High-resolution figure and video files: See our detailed guidelines for preparing your production-ready images, <http://jcb.rupress.org/fig-vid-guidelines>.

-- Cover images: If you have any striking images related to this story, we would be happy to

consider them for inclusion on the journal cover. Submitted images may also be chosen for highlighting on the journal table of contents or JCB homepage carousel. Images should be uploaded as TIFF or EPS files and must be at least 300 dpi resolution.

****It is JCB policy that if requested, original data images must be made available to the editors. Failure to provide original images upon request will result in unavoidable delays in publication. Please ensure that you have access to all original data images prior to final submission.****

****The license to publish form must be signed before your manuscript can be sent to production. A link to the electronic license to publish form will be sent to the corresponding author only. Please take a moment to check your funder requirements before choosing the appropriate license.****

Thank you for this interesting contribution, we look forward to publishing your paper in the Journal of Cell Biology.

Sincerely,

William Prinz, PhD
Monitoring Editor, Journal of Cell Biology

Melina Casadio, PhD
Senior Scientific Editor, Journal of Cell Biology

Reviewer #1 (Comments to the Authors (Required)):

The authors have appropriately answered my criticisms and seem to go a long way to answer the other reviewers. I am comfortable with the changes made in the paper. I have no new critical comments.

Reviewer #2 (Comments to the Authors (Required)):

As indicated in my initial review the manuscript by Du et al describes an intriguing finding that ORP5 localizes to the ER-LD contacts to deliver PS at the expense of PI4P, and most importantly for the first time they show that PI4P is present on LDs. The authors have carried out an extensive revision of the manuscript and they have addressed all my concerns for improving the paper.

Reviewer #3 (Comments to the Authors (Required)):

The authors have addressed the concerns satisfactorily. However, related to question #9, I think they missed the point about ER marker isolation with LDs (Fig. 3D). If the model is correct, that

ORP5 is tethered to the LD by the OHD and to the ER by the TM domain, one would conclude that any ORP5 isolated on a LD would have its TM domain still imbedded in the ER and as a result there would be a ER protein signature on the LD. Since they could detect calnexin, the conclusion is either ORP5 is in a unique ER domain with no calnexin or the ER membrane is not there. The last point is a possibility since the C-terminus of ORP5 looks a bit like a tail anchored protein that is embedded in this target membrane (ER) post-translationally. This raises the issue that overexpression of ORP5 might have saturated ORP5 targeting sites on the ER and ends up on LDs instead.